# Memory, Benchmark & Robots: A Benchmark for Solving Complex Tasks with Reinforcement Learning

**Egor Cherepanov**[1,2]  **Nikita Kachaev**[1,3]  **Alexey K. Kovalev**[1,2]  **Aleksandr I. Panov**[1,2]

[1]AIRI, Moscow, Russia  [2]MIPT, Dolgoprudny, Russia  [3]HSE University, Moscow, Russia

{cherepanov,kachaev,kovalev,panov}@airi.net

## Abstract

Memory is crucial for enabling agents to tackle complex tasks with temporal and spatial dependencies. While many reinforcement learning (RL) algorithms incorporate memory, the field lacks a universal benchmark to assess an agent's memory capabilities across diverse scenarios. This gap is particularly evident in tabletop robotic manipulation, where memory is essential for solving tasks with partial observability and ensuring robust performance, yet no standardized benchmarks exist. To address this, we introduce **MIKASA** (**M**emory-**I**ntensive **Sk**ills **A**ssessment **S**uite for **A**gents), a comprehensive benchmark for memory RL, with three key contributions: (1) we propose a comprehensive classification framework for memory-intensive RL tasks, (2) we collect **MIKASA-Base** – a unified benchmark that enables systematic evaluation of memory-enhanced agents across diverse scenarios, and (3) we develop **MIKASA-Robo**[1] – a novel benchmark of 32 carefully designed memory-intensive tasks that assess memory capabilities in tabletop robotic manipulation. Our work introduces a unified framework to advance memory RL research, enabling more robust systems for real-world use. **MIKASA is available at** **https://tinyurl.com/membenchrobots**.

## 1 Introduction

Many real-world problems involve partial observability [43], where an agent lacks full access to the environment's state. These tasks often include sequential decision-making [8], delayed or sparse rewards, and long-term information retention [54, 72]. One approach to tackling these challenges is to equip the agent with memory, allowing it to utilize historical information [64, 67]. While there are well-established benchmarks in Natural Language Processing [3, 5], the evaluation of memory in reinforcement learning (RL) remains fragmented. Existing benchmarks, such as POPGym [65], DMLab-30 [39] and Memory-Gym [75], focus on specific aspects of memory utilization, as they are designed around particular problem domains.

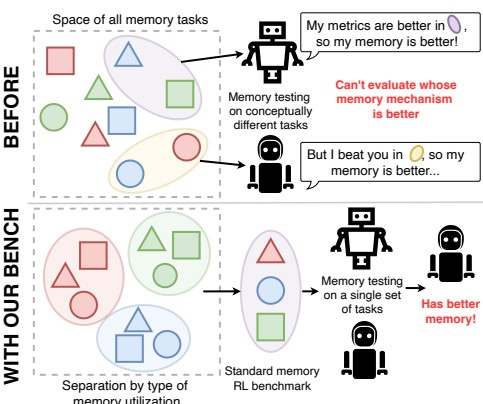

Figure 1: Systematic classification of problems with memory in RL reveals distinct memory utilization patterns and enables objective evaluation of memory mechanisms across different agents.

In contrast to classical RL, where benchmarks like Atari [7] and MuJoCo [89] serve as universal standards, memory-enhanced agents are typically evaluated on custom environments developed along-

---

[1]pip install mikasa-robo-suite

Submitted to 39th Conference on Neural Information Processing Systems (NeurIPS 2025). Do not distribute.

Table 1: **MIKASA-Robo**: A benchmark comprising **32 memory-intensive robotic manipulation tasks** across 12 categories. Each task varies in difficulty and configuration modes. The table specifies episode timeout (T), the necessary information that the agent must memorize in order to succeed (Oracle Info), and task instructions (Prompt) for each environment. See Appendix H for details.

| Memory Task | Mode | Brief description of the task | T | Oracle Info | Prompt | Memory |
|---|---|---|---|---|---|---|
| ShellGame | Touch Push Pick | Memorize the position of the ball after some time being covered by the cups and then interact with the cup the ball is under | 90 | cup_with_ball_number | — | Object |
| Intercept | Slow Medium Fast | Memorize the positions of the rolling ball, estimate its velocity through those positions, and then aim the ball at the target | 90 | initial_velocity | — | Spatial |
| InterceptGrab | Slow Medium Fast | Memorize the positions of the rolling ball, estimate its velocity through those positions, and then catch the ball with the gripper and lift it up | 90 | initial_velocity | — | Spatial |
| RotateLenient | Pos PosNeg | Memorize the initial position of the peg and rotate it by a given angle | 90 | y_angle_diff | target_angle | Spatial |
| RotateStrict | Pos PosNeg | Memorize the initial position of the peg and rotate it to a given angle without shifting its center | 90 | y_angle_diff | target_angle | Spatial |
| TakeItBack-v0 | — | Memorize the initial position of the cube, move it to the target region, and then return it to its initial position | 180 | xyz_initial | — | Spatial |
| RememberColor | 3\5\9 | Memorize the color of the cube and choose among other colors | 60 | true_color_indices | — | Object |
| RememberShape | 3\5\9 | Memorize the shape of the cube and choose among other shapes | 60 | true_shape_indices | — | Object |
| RememberShape-AndColor | 3×2\3×3\5×3 | Memorize the shape and color of the cube and choose among other shapes and colors | 60 | true_shapes_info true_colors_info | — | Object |
| BunchOfColors | 3\5\7 | Remember the colors of the set of cubes shown simultaneously in the bunch and touch them in any order | 120 | true_color_indices | — | Capacity |
| SeqOfColors | 3\5\7 | Remember the colors of the set of cubes shown sequentially and then select them in any order | 120 | true_color_indices | — | Capacity |
| ChainOfColors | 3\5\7 | Remember the colors of the set of cubes shown sequentially and then select them in the same order | 120 | true_color_indices | — | Sequential |
| **Total: 32 tabletop robotic manipulation memory-intensive tasks in 12 groups** | | | | | | |

side their proposals Table 2. This fragmented evaluation landscape obscures important performance variations across different memory tasks. For instance, an agent might excel at maintaining object attributes over extended periods while struggling with sequential recall challenges. Such task-specific strengths and limitations often remain hidden due to narrow evaluation scopes, underscoring the need for a comprehensive benchmark that spans diverse memory-intensive scenarios.

The challenge of memory evaluation becomes particularly evident in robotics. While some robotic tasks naturally involve partial observability, e.g. navigation tasks [2, 94], many studies artificially create partially observable scenarios from Markov Decision Processes (MDPs) [42] by introducing observation noise or masking parts of the state space [52, 55, 64, 85]. However, these approaches do not fully capture the complexity of real-world robotic challenges [55], where tasks may require the agent to recall past object configurations, manipulate occluded objects, or perform multi-step procedures that depend heavily on memory. Such tasks include, for example, situations where a service robot needs to memorize occluded objects (e.g., a plate hidden under a towel) or where a home robot needs to accurately wipe the door of a microwave oven several times. Without memory, the robot wouldn't detect the plate in the first case, and in the second, it would wipe the door endlessly, unsure whether it has cleaned the area or if it's time to stop.

In this paper, we aim to address these challenges with the following four contributions:

1. **Memory Tasks Classification.** We propose a simple yet comprehensive framework that organizes memory-intensive tasks into four key categories. This structure enables systematic evaluation without added complexity (Figure 1), offering a clear guide for selecting environments that reflect core memory challenges in RL and robotics (Section 4).

2. **Memory-RL Benchmark.** We introduce **MIKASA-Base**, a Gymnasium-based [90] framework for evaluating memory-enhanced RL agents (Section 5).

3. **Robotic Manipulation Tasks.** We introduce **MIKASA-Robo**, a suite of 32 robotic tasks targeting specific memory-dependent skills in realistic settings (Section 6), and evaluate them using popular Online RL baselines (Subsection 6.2) and Visual-Language-Action (VLA) models (Subsection 6.4).

4. **Robotic Manipulation Datasets.** We release datasets for all 32 MIKASA-Robo memory-intensive tasks to support Offline RL research (see Appendix B), and conduct extensive evaluations using a range of Offline RL baselines (Subsection 6.3).

## 2   Related Works

Multiple RL benchmarks are designed to assess agents' memory capabilities. DMLab-30 [39] provides 3D navigation and puzzle tasks, focusing on long-horizon exploration and spatial recall.

PsychLab [56] extends DMLab by incorporating tasks that probe cognitive processes, including working memory. MiniGrid and MiniWorld [12] emphasize partial observability in lightweight 2D and 3D environments, while MiniHack [78] builds on NetHack [53], offering small rogue-like scenarios that require both short- and long-term memory. BabyAI [11] combines natural language instructions with grid-based tasks, requiring memory for multi-step command execution. POPGym [65] standardizes memory evaluation with tasks ranging from pattern-matching puzzles to complex sequential decision-making. BSuite [70] offers a suite of carefully designed experiments that test core RL capabilities, including memory, through controlled tasks on exploration, credit assignment, and scalability. Memory Gym [75] offers a suite of 2D grid environments with partial observability, designed to benchmark memory capabilities in decision-making agents, including endless versions of tasks for evaluating memory over extremely long time intervals. Memory Maze [73] presents 3D maze navigation tasks that require memory to solve efficiently.

Table 2: Key memory-intensive environments from the reviewed studies for evaluating agent memory. The Atari [7] environment with frame stacking is included to illustrate that many memory-enhanced agents are tested solely in MDP. Benchmark first introduced in the same work . Benchmark is open-sourced.

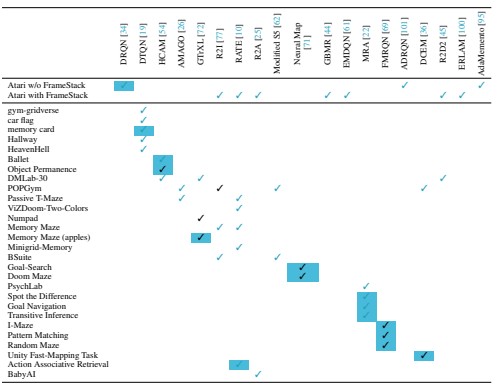

While these benchmarks offer valuable insights into memory mechanisms, they generally focus on abstract puzzles or navigation tasks. However, none of them fully encompass the broad range of memory utilization scenarios an agent may encounter, and the tasks themselves often differ fundamentally across benchmarks, making direct comparison of memory-enhanced agents difficult. In the robotics domain, memory requirements become particularly challenging due to the physical nature of manipulation tasks. Unlike abstract environments, robotic manipulation involves complex physical interactions and multi-step procedures demanding both spatial and temporal memory. Existing memory-intensive benchmarks, while useful for diagnostic purposes, struggle to capture these domain-specific challenges. The physical control and object interaction inherent in manipulation tasks introduce additional complexities not addressed by traditional memory evaluation frameworks.

Efforts have been made to classify memory-intensive environments by specific attributes. For example, Ni et al. [68] divides them into memory/credit assignment based on temporal horizons. Yue et al. [97] proposes memory dependency pairs to model how past events influence current decisions, aiding imitation learning in partially observable tasks. Cherepanov et al. [9] defines agent memory types: long-term vs. short-term (based on context length), and declarative vs. procedural (based on environments and episodes), and formalizes memory-intensive environments. Leibo et al. [56] instead adapts tasks from cognitive psychology and psychophysics to evaluate agents on human cognitive benchmarks. While these classifications highlight aspects of memory, they overlook physical dimensions in robotics. The link between physical interaction and memory remains underexplored, motivating a framework for spatio-temporal memory in real-world tasks.

Concurrent with our work Fang et al. [21] also proposed MemoryBench, a benchmark for memory-intensive manipulation consisting of only three tasks designed to access only one type of memory, spatial memory. This benchmark is based on RLBench [40], which does not allow efficient parallelization of training.

# 3 Background

## 3.1 Partially Observable Markov Decision Process

Partially Observable Markov Decision Process (POMDP) [42] extend MDP to account for partial observability, where an agent observes only noisy or incomplete information about the true environments state. POMDP defined by a tuple $(S, A, T, R, \Omega, O, \gamma)$, where: $S$ is the set of states representing the complete environment configuration; $A$ is the action space; $T(s'|s, a) : S \times A \times S \to [0, 1]$ is

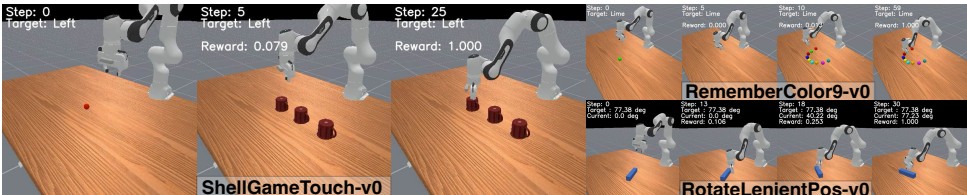

Figure 2: Illustration of demonstrative memory-intensive tasks execution from the proposed MIKASA-Robo benchmark. The `ShellGameTouch-v0` task requires the agent to memorize the ball's location under mugs and touch the correct one. In `RememberColor9-v0`, the agent must memorize a cube's color and later select the matching one. In `RotateLenientPos-v0`, the agent must rotate a peg while keeping track of its previous rotations.

the transition function defining the probability of reaching state $s'$ from state $s$ after taking action $a$; $R(s, a) : S \times A \to \mathbb{R}$ is the reward function specifying the immediate reward for taking action $a$ in state $s$; $\Omega$ is the observation space containing all possible observations; $O(o|s, a) : S \times A \times \Omega \to [0, 1]$ is the observation function defining the probability of observing $o$ after taking action $a$ and reaching state $s$; $\gamma \in [0, 1)$ is the discount factor determining the importance of future rewards. The objective is to find a policy $\pi$ that maximizes the expected discounted cumulative reward: $\mathbb{E}_\pi \left[ \sum_{t=0}^{\infty} \gamma^t R(s_t, a_t) \right]$, where $a_t \sim \pi(\cdot | o_{1:t})$ depends on the history of observations rather than the true state. Relying on partial observations makes POMDPs harder to solve than MDPs.

### 3.2 Memory-intensive environments

Memory-intensive environment is an environment where agents must leverage past experiences to make decisions, often in problems with long-term dependencies or delayed rewards. More formally, following Cherepanov et al. [9], a memory-intensive task is a POMDP where there exists a correlation horizon $\xi > 1$, representing the minimum number of timesteps between an event critical for decision-making and when that information must be recalled. Popular memory-intensive environments in RL are listed in Table 2. One way to solving memory-intensive environments is to augment agents with memory mechanisms (see Appendix E).

### 3.3 Robotic Tabletop Manipulation

Robotic tabletop manipulation [80] involves robots manipulating objects on flat surfaces through actions like grasping, pushing, and picking. While crucial for real-world applications [57], most existing simulators treat these tasks as MDPs without memory requirements, failing to capture the spatio-temporal dependencies present in real scenarios. This limitation hinders the development of memory-enhanced agents for practical applications.

## 4 Classification of memory-intensive tasks

The evaluation of memory capabilities in RL faces two major challenges. First, as shown in Table 2, research studies use different sets of environments with minimal overlap, making it difficult to compare memory-enhanced agents across studies. Second, even within individual studies, benchmarks may focus on testing similar memory aspects (e.g., remembering object locations) while neglecting others (e.g., reconstructing sequential events), leading to incomplete evaluation of agents' memory.

Different architectures may exhibit varying performance across memory tasks. For instance, an architecture optimized for long-term object property recall might struggle with sequential memory tasks, yet these limitations often remain undetected due to the narrow focus of existing evaluation approaches.

To address these challenges, we propose a systematic approach to memory evaluation in RL. Drawing from established research in developmental psychology and cognitive science, where similar memory challenges have been extensively studied in humans, we develop a categorization framework consisting of four distinct memory task classes, detailed in Subsection 4.2.

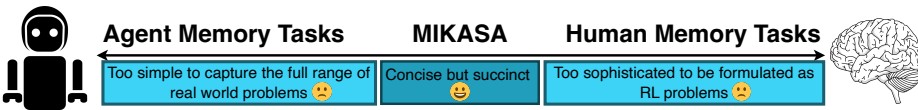

Figure 3: MIKASA bridges the gap between human-like memory complexity and RL agents requirements. While agents tasks don't require the full spectrum of human memory capabilities, they can't be reduced to simple spatio-temporal dependencies. MIKASA provides a balanced framework that captures essential memory aspects for agents tasks while maintaining practical simplicity.

## 4.1 Memory: From Cognitive Science to RL

In developmental psychology and cognitive science, memory is classified into categories based on cognitive processes. Key concepts include object permanence [74], which involves remembering the existence of objects out of sight, and categorical perception [60], where objects are grouped based on attributes like color or shape. Working memory [4] and memory span [16] refer to the ability to hold and manipulate information over time, while causal reasoning [50] and transitive inference [35] involve understanding cause-and-effect relationships and deducing hidden relationships, respectively.

The RL field has attempted to utilize these concepts in the design of specific memory-intensive environments [22, 54], but these have been limited at the task design level. Of particular interest, however, is how existing memory-intensive tasks can be categorized using these concepts to develop a benchmark on which to test the greatest number of memory capabilities of memory-enhanced agents, and it is this problem that we address in this paper. Thus, we aim to provide a balanced framework that covers important aspects of memory for real-world applications while maintaining practical simplicity (see Figure 3).

## 4.2 Taxonomy of Memory Tasks

We introduce a comprehensive task classification framework for evaluating memory mechanisms in RL. Our framework categorizes memory-intensive tasks into four fundamental types, each targeting distinct aspects of memory capabilities:

1. **Object Memory.** Tasks that evaluate an agent's ability to maintain object-related information over time, particularly when objects become temporarily unobservable. These tasks align with the cognitive concept of object permanence, requiring agents to track object properties when occluded, maintain object state representations, and recognize encountered objects. Example: a robot remembers which fruit it put in the fridge.

2. **Spatial Memory.** Tasks focused on environmental awareness and navigation, where agents must remember object locations, maintain mental maps of environment layouts, and navigate based on previously observed spatial information. Example: the robot remembers the position of a mug it moved while cleaning and returns it to its place.

3. **Sequential Memory.** Tasks that test an agent's ability to process and utilize temporally ordered information, similar to human serial recall and working memory. These tasks require remembering action sequences, maintaining order-dependent information, and using past decisions to inform future actions. Example: a robot memorizes the order of the ingredients it has added to a soup.

4. **Memory Capacity.** Tasks that challenge an agent's ability to manage multiple pieces of information simultaneously, analogous to human memory span. These tasks evaluate information retention limits and multi-task information processing. Example: a robot is able to memorize the positions of several different objects while cleaning a table.

This classification framework enables systematic evaluation of memory-enhanced RL agents across diverse scenarios. By providing a structured approach to memory task categorization, we establish a foundation for comprehensive benchmarking that spans the wide spectrum of memory requirements. In the following section, we present a carefully curated set of tasks based on this classification, forming the basis of our proposed MIKASA benchmark.

## 5 MIKASA-Base

**Motivation and Overview.** Despite the importance of memory in decision-making, the RL community lacks standardized tools for benchmarking memory capabilities. Existing studies typically introduce bespoke environments tailored to their proposed algorithms, leading to fragmentation and limited comparability across works (see Table 2). Moreover, many popular memory benchmarks focus narrowly on specific memory types, overlooking the diversity of memory demands found in real-world applications. To address this gap, we introduce **MIKASA-Base**, a unified benchmark that consolidates widely used open-source memory-intensive environments under a common Gym-like API. Our goal is to streamline reproducibility, support fair comparisons, and promote systematic evaluation of memory in RL.

**Benchmark Design Principles.** MIKASA-Base is designed around core principles that support rigorous and interpretable evaluation of memory in RL. To disentangle memory from unrelated challenges, we organize tasks into two tiers. The first tier consists of **diagnostic** vector-

Table 3: Analysis of established robotics frameworks with manipulation tasks, comparing their support for memory-intensive tasks. † – excluding Franka Kitchen. ∗ – concurrent work with three memory tasks with only one type of memory.

| Robotics Framework with Manipulation Tasks | Memory Tasks | | |
| --- | --- | --- | --- |
| | Manipulation | Atomic | Low-level actions |
| MIKASA-Robo (**Ours**) | ✓ | ✓ | ✓ |
| MemoryBench∗ [21] | ✓ | ✓ | ✓ |
| ManiSkill3 [87] | ✗ | ✗ | ✗ |
| ManiSkill-HAB [81] | ✗ | ✗ | ✗ |
| FetchBench [32] | ✗ | ✗ | ✗ |
| RoboCasa [66] | ✗ | ✗ | ✗ |
| Gymnasium-Robotics† [17] | ✗ | ✗ | ✗ |
| BEHAVIOR-1K [59] | ✓ | ✗ | ✗ |
| ARNOLD [24] | ✗ | ✗ | ✗ |
| iGibson 2.0 [58] | ✓ | ✗ | ✗ |
| VIMA [41] | ✓ | ✓ | ✗ |
| Isaac Sim [63] | ✗ | ✗ | ✗ |
| panda-gym [23] | ✗ | ✗ | ✗ |
| Habitat 2.0 [86] | ✗ | ✗ | ✗ |
| Meta-World [96] | ✗ | ✗ | ✗ |
| CausalWorld [1] | ✗ | ✗ | ✗ |
| RLBench [40] | ✗ | ✗ | ✗ |
| robosuite [102] | ✗ | ✗ | ✗ |
| dm_control [91] | ✗ | ✗ | ✗ |
| Franka Kitchen [29] | ✗ | ✗ | ✗ |
| SURREAL [20] | ✗ | ✗ | ✗ |
| AI2-THOR [49] | ✗ | ✗ | ✗ |

based environments that isolate specific memory mechanisms. The second tier includes **complex** image-based tasks that incorporate realistic perception challenges, thus more closely resembling real-world settings. This hierarchical structure enables researchers to validate memory capabilities incrementally – from atomic reasoning to high-dimensional sensory input.

**Task Classification and Selection.** Building on our taxonomy from Subsection 4.2, we systematically reviewed open-source memory benchmarks and categorized their tasks into four distinct types of memory usage. We selected a diverse yet representative subset of environments to cover this taxonomy – ranging from object permanence to sequential planning. All selected tasks are unified under a single, consistent API. Descriptions are provided in Appendix I, and an overview of MIKASA-Base tasks appears in Table 6. This consolidation supports architectural ablations, direct comparison of methods, and simplified evaluation pipelines. Implementation details can be found in Appendix C.

MIKASA-Base provides the first systematic and unified benchmark for evaluating memory in RL. It mitigates fragmentation by standardizing task access and evaluation, and its structured progression enables precise attribution of memory-related agent failures. By covering a broad spectrum of memory challenges within a common framework, MIKASA-Base offers a foundation for robust, reproducible research in memory-centric RL.

## 6 MIKASA-Robo

The landscape of robotic manipulation frameworks reveals significant limitations in addressing memory-intensive tasks. While partial observability is well-studied in navigation, manipulation scenarios are still predominantly evaluated under full observability, with limited focus on memory demands (see Table 3). Among frameworks that do consider memory, BEHAVIOR-1k [59] and iGibson 2.0 [58] include highly complex, non-atomic tasks, which obscure the evaluation of specific memory mechanisms. VIMA [41] relies on high-level action abstractions, limiting temporal memory assessment. To address these gaps, we introduce **MIKASA-Robo**, a benchmark specifically designed to evaluate diverse memory skills in robotic manipulation through well-isolated, fine-grained tasks.

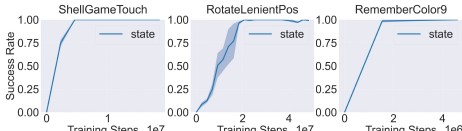
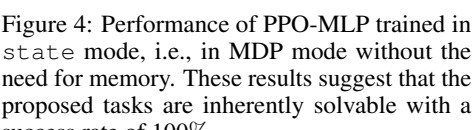

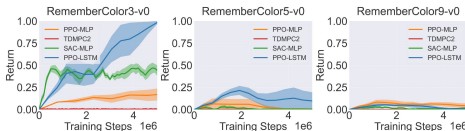

Figure 4: Performance of PPO-MLP trained in `state` mode, i.e., in MDP mode without the need for memory. These results suggest that the proposed tasks are inherently solvable with a success rate of 100%.

Figure 5: Online RL baselines with MLP and LSTM backbones trained in `RGB+joints` mode on the `RememberColor-v0` environment with dense rewards. Both architectures fail to solve medium and high complexity tasks.

Concurrently with our work, Fang et al. [21] proposed **MemoryBench**, a benchmark focused on spatial memory with three robotic tasks. In contrast, MIKASA-Robo spans four memory categories and 32 tasks, enabling broader and more systematic evaluation of memory mechanisms in RL agents.

**MIKASA-Robo** is a benchmark designed for memory-intensive robotic tabletop manipulation tasks, simulating real-world challenges commonly encountered by robots. These tasks include locating occluded objects, recalling previous configurations, and executing complex sequences of actions over extended time horizons. By incorporating meaningful partial observability, this framework offers a systematic approach to test an agent's memory mechanisms.

Building upon the robust foundation of ManiSkill3 framework [87], our benchmark leverages its efficient parallel GPU-based training capabilities to create and evaluate these tasks.

### 6.1 MIKASA-Robo Manifestation

In designing the tasks, we drew inspiration from the four memory types identified in our classification framework (Subsection 4.2). We developed **32 tasks across 12 categories of robotic tabletop manipulation**, each targeting specific aspects of object memory, spatial memory, sequential memory, and memory capacity. These tasks feature varying levels of complexity, allowing for systematic evaluation of different memory mechanisms. For instance, some tasks test object permanence by requiring the agent to track occluded objects, while others challenge sequential memory by requiring the reproduction of a strict order of actions. A summary of these tasks and their corresponding memory types is provided in Table 1, with detailed descriptions in Appendix H.

To illustrate the concept of our memory-intensive framework, we present `ShellGameTouch-v0`, `RememberColor-v0`, and `RotateLenientPos-v0` tasks in Figure 2. In the `ShellGameTouch-v0` task, the agent observes a red ball placed in one of three positions over the first 5 steps ($t \in [0, 4]$). At $t = 5$, the ball and the three positions are covered by mugs. The agent must then determine the location of the ball by interacting with the correct mug. In the simplest mode (`Touch`), the agent only needs to touch the correct mug, whereas in other modes, it must either push or lift the mug. In the `RememberColor-v0` task, the agent observes a cube of a specific color for 5 steps ($t \in [0, 4]$). After the cube disappears for 5 steps, 3, 5, or 9 (depending on task mode) cubes of different colors appear at $t = 10$. The agent's task is to identify and select the same cube it initially saw. In the `RotateLenientPos-v0` task, the agent must rotate a randomly oriented peg by a specified clockwise angle.

The MIKASA-Robo benchmark offers multiple training modes: `state` (complete vector information including oracle data and Tool Center Point (TCP) pose), `RGB` (top-view and gripper-camera images with TCP position), `joints` (joint states and TCP pose), `oracle` (task-specific environment data for debugging), and `prompt` (static task instructions). While any mode combination is possible, **RGB+joints serves as the standard memory testing configuration**, with `state` mode reserved for MDP-based tasks.

The MIKASA-Robo benchmark implements two types of reward functions: dense and sparse. The dense reward provides continuous feedback based on the agent's progress towards the goal, while the sparse reward only signals task completion. While dense rewards facilitate faster learning in our experiments, sparse rewards better reflect real-world scenarios where intermediate feedback is often unavailable, making them crucial for evaluating practical applicability of memory-enhanced agents.

## 6.2 Online RL baselines

For the experimental evaluation, we chose on-policy Proximal Policy Optimization (PPO, [79]) with two underlying architectures: Multilayer Perceptron (MLP) and Long Short-Term Memory (LSTM, [37]), as well as popular in robotics off-policy Soft Actor-Critic (SAC, [30]) and model-based Temporal Difference Learning for Model Predictive Control (TD-MPC2, [33]).

The MLP variant serves as a memory-less baseline, while LSTM represents a widely-adopted memory mechanism in RL, known for its effectiveness in solving POMDPs [67]. This choice of architectures enables direct comparison between memory-less and memory-enhanced agents while validating our benchmark's ability to as-

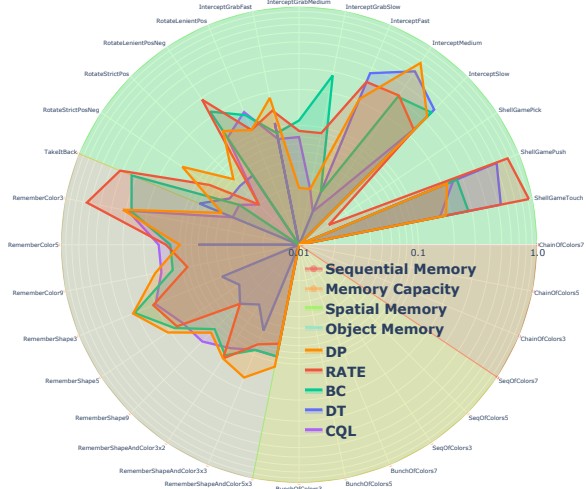

Figure 6: Results of Offline RL baselines with memory (RATE, DT) and without memory (BC-MLP, CQL-MLP, DP) on all 32 MIKASA-Robo tasks. Training was performed in `RGB` mode with sparse rewards (success condition).

sess memory. We focus specifically on these fundamental architectures as they align with our primary goal of benchmark validation rather than comprehensive algorithm comparison. To demonstrate that all proposed environments are solvable with 100% success rate (SR), we trained a PPO-MLP agent using `state` mode, where it had full access to system information. Results for select tasks are shown in Figure 4; full results are in Appendix F.

Training under the `RGB+joints` mode with dense rewards reveals the memory-intensive nature of our tasks. Using the `RememberColor-v0` task as an example, PPO-LSTM demonstrates superior performance compared to PPO-MLP when distinguishing between three colors (see Figure 5). However, both agents' success rates drop dramatically to near-zero as the task complexity increases to five or nine colors. Moreover, under sparse reward conditions, both architectures fail to solve even the three-color variant (see Appendix F, Figure 10). Additionally, our findings indicate that, while SAC and TD-MPC2 exhibit higher sample efficiency compared to PPO-MLP, when faced with more complex challenges, the lack of an explicit memory mechanism becomes a critical shortcoming, resulting in low performance, which also emphasizes the inappropriateness of algorithms common in the robotics community for memory-intensive tasks. These results validate our benchmark's effectiveness in evaluating agents' memory, showing clear performance degradation as memory demands increase.

## 6.3 Offline RL baselines

Since dense rewards are typically not available in the real world, it is of particular interest to train on sparse rewards represented as a binary flag of a successfully completed episode. Whereas models with online learning are extremely hard to handle in this setting, we also conducted experiments with five Offline RL models: Decision Transformer (DT) [8] and Recurrent Action Transformer with Memory (RATE) [10]) based on the Transformer architecture, Standard Behavioral Cloning (BC) and Conservative Q-Learning (CQL) [51]) with MLP backbones, as well as Diffusion Policy (DP) [13]) – a recent and popular approach in robotic manipulation that leverages diffusion models for direct action prediction.

Experimental results with Offline RL models trained using two RGB camera views and sparse rewards are presented in Figure 6. As can be seen from Figure 6, none of the models – including those explicitly designed for sequence modeling – were able to successfully solve the majority of MIKASA-Robo tasks, demonstrating the challenge posed by the benchmark. Training was conducted using datasets consisting of 1000 successful trajectories per task (see Appendix B for details).

Notably, none of the evaluated models were able to solve tasks requiring high Memory Capacity or Sequential Memory, further underscoring their complexity. More detailed results for Offline RL algorithms are presented in Appendix, Table 5.

Table 4: Performance of VLA models on selected memory-intensive tasks from the MIKASA-Robo benchmark. Reported values denote average success rates over 100 evaluation episodes (mean ± sem). Tasks include spatial reasoning (`ShellGameTouch`, `InterceptMedium`) and color-based memory retrieval (`RememberColor3/5/9`).

| Model | ShellGameTouch- | InterceptMedium | RememberColor3 | RememberColor5 | RememberColor9 |
|---|---|---|---|---|---|
| **Octo-small** | 0.46 ± 0.05 | 0.39 ± 0.04 | 0.45 ± 0.06 | 0.17 ± 0.03 | 0.11 ± 0.03 |
| **OpenVLA** ($K$=4) | 0.12 ± 0.05 | 0.06 ± 0.02 | 0.21 ± 0.00 | 0.09 ± 0.02 | 0.08 ± 0.02 |
| **OpenVLA** ($K$=8) | 0.47 ± 0.05 | 0.14 ± 0.03 | 0.59 ± 0.04 | 0.16 ± 0.03 | 0.06 ± 0.02 |

## 6.4 VLA baselines

To investigate the capabilities of state-of-the-art Visual-Language-Action (VLA) models in memory-intensive robotic tasks, we selected two representative baselines: Octo [88] and OpenVLA [47]. Although neither model explicitly claims to implement sophisticated memory mechanisms, these experiments provide valuable insights into the current state of memory capabilities in VLA agents.

Octo is a transformer with diffusion heads trained from scratch on 25 Open X-Embodiment datasets [15]; in our experiments, only the readout heads were fine-tuned using the full pretrained context length of 10 and action chunk size (K=4). OpenVLA uses a Prismatic-7B backbone [46], fine-tuned for action prediction with LoRA adapters, action chunking, and an $L_1$ loss [48]. We tested chunk sizes K=4 and K=8. Both models were trained on 250 expert trajectories per task, using $128 \times 128$ RGB image pairs (base and wrist views) and end-effector control (see Appendix D).

Experimental results (Table 4) reveal notable trends. Octo (context size 10) outperforms random on simpler tasks, suggesting some innate memory capacity, but its performance degrades with task complexity, indicating limited scalability. OpenVLA shows contrasting behavior across chunk sizes: with $K = 8$, it exceeds random on tasks like `RememberColor3` and `ShellGameTouch`, despite lacking step-wise history. However, performance drops sharply on harder tasks. With $K = 4$, it fails across the board. These results suggest that larger chunk sizes can help bypass explicit memory by generating full trajectories from early cues, but this strategy fails with smaller chunks, where initial correct actions often give way to confusion. Thus, action chunking offers limited compensation for the absence of a true memory mechanism.

The sharp decline in performance on higher-complexity tasks underscores the necessity for dedicated memory architectures and validates the importance of the multi-difficulty hierarchy in MIKASA-Robo to prevent such "shortcuts". Our experiments with Octo and OpenVLA highlights a critical gap in current VLA models: the absence of effective long-term memory leads to brittle performance on tasks demanding strong memory capabilities. These experiments not only illuminate present limitations but also reinforce the value of the MIKASA-Robo benchmark.

## 7 Limitations

While our benchmark provides a comprehensive evaluation framework, some limitations remain. In particular, the performance of Octo and OpenVLA may not reflect their full potential, as we performed limited fine-tuning due to computational constraints. Future work could explore more extensive adaptation of large VLA models within MIKASA to better assess their memory capabilities. Additionally, while MIKASA covers a broad range of memory challenges, further extensions could incorporate tasks with longer temporal dependencies or meta-RL.

## 8 Conclusion

We present **MIKASA**, a unified benchmark suite for evaluating memory in RL. Our work addresses key gaps in the field by introducing: (1) a taxonomy of memory types – object, spatial, sequential, and capacity; (2) **MIKASA-Base**, a standardized collection of open-source memory tasks; (3) **MIKASA-Robo**, a suite of 32 robotic manipulation tasks targeting diverse memory demands; and (4) accompanying offline datasets to support reproducible evaluation. Experiments with online, offline, and VLA agents reveal that current methods struggle with many tasks, highlighting the need for better memory architectures. MIKASA aims to guide and accelerate progress in memory-intensive RL for real-world applications. The **MIKASA-Robo** suite is publicly available and can be easily installed via `pip install mikasa-robo-suite`.

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

# NeurIPS Paper Checklist

1. **Claims**

   Question: Do the main claims made in the abstract and introduction accurately reflect the paper's contributions and scope?

   Answer: [Yes]

   Justification: The abstract and introduction clearly state the paper's four main contributions: (1) a taxonomy of memory task types in RL, (2) the MIKASA-Base benchmark unifying open-source memory-intensive tasks, (3) the MIKASA-Robo benchmark with 32 robotic tasks targeting diverse memory skills. These claims are substantiated by theoretical motivation and extensive experimental validation with online RL Subsection 6.2, offline RL Subsection 6.3, and VLA models Subsection 6.4. Limitations and assumptions are transparently discussed.

   Guidelines:

   - The answer NA means that the abstract and introduction do not include the claims made in the paper.
   - The abstract and/or introduction should clearly state the claims made, including the contributions made in the paper and important assumptions and limitations. A No or NA answer to this question will not be perceived well by the reviewers.
   - The claims made should match theoretical and experimental results, and reflect how much the results can be expected to generalize to other settings.
   - It is fine to include aspirational goals as motivation as long as it is clear that these goals are not attained by the paper.

2. **Limitations**

   Question: Does the paper discuss the limitations of the work performed by the authors?

   Answer: [Yes]

   Justification: The paper includes a dedicated Limitations section (Section 7), where the authors acknowledge that the evaluation of Octo and OpenVLA may not fully reflect the models' capabilities due to limited fine-tuning constrained by computational resources. The section also suggests directions for extending the benchmark to cover more complex settings. This discussion transparently outlines the scope of claims and experimental coverage without undermining the paper's contributions.

   Guidelines:

   - The answer NA means that the paper has no limitation while the answer No means that the paper has limitations, but those are not discussed in the paper.
   - The authors are encouraged to create a separate "Limitations" section in their paper.
   - The paper should point out any strong assumptions and how robust the results are to violations of these assumptions (e.g., independence assumptions, noiseless settings, model well-specification, asymptotic approximations only holding locally). The authors should reflect on how these assumptions might be violated in practice and what the implications would be.
   - The authors should reflect on the scope of the claims made, e.g., if the approach was only tested on a few datasets or with a few runs. In general, empirical results often depend on implicit assumptions, which should be articulated.
   - The authors should reflect on the factors that influence the performance of the approach. For example, a facial recognition algorithm may perform poorly when image resolution is low or images are taken in low lighting. Or a speech-to-text system might not be used reliably to provide closed captions for online lectures because it fails to handle technical jargon.
   - The authors should discuss the computational efficiency of the proposed algorithms and how they scale with dataset size.
   - If applicable, the authors should discuss possible limitations of their approach to address problems of privacy and fairness.

- While the authors might fear that complete honesty about limitations might be used by reviewers as grounds for rejection, a worse outcome might be that reviewers discover limitations that aren't acknowledged in the paper. The authors should use their best judgment and recognize that individual actions in favor of transparency play an important role in developing norms that preserve the integrity of the community. Reviewers will be specifically instructed to not penalize honesty concerning limitations.

3. **Theory assumptions and proofs**

Question: For each theoretical result, does the paper provide the full set of assumptions and a complete (and correct) proof?

Answer: [NA]

Justification: The paper does not include formal theoretical results or proofs. While Section 4 presents a conceptual taxonomy of memory tasks grounded in cognitive science, it is not formalized as a mathematical theory with theorems or proofs. Therefore, this question is not applicable.

Guidelines:

- The answer NA means that the paper does not include theoretical results.
- All the theorems, formulas, and proofs in the paper should be numbered and cross-referenced.
- All assumptions should be clearly stated or referenced in the statement of any theorems.
- The proofs can either appear in the main paper or the supplemental material, but if they appear in the supplemental material, the authors are encouraged to provide a short proof sketch to provide intuition.
- Inversely, any informal proof provided in the core of the paper should be complemented by formal proofs provided in appendix or supplemental material.
- Theorems and Lemmas that the proof relies upon should be properly referenced.

4. **Experimental result reproducibility**

Question: Does the paper fully disclose all the information needed to reproduce the main experimental results of the paper to the extent that it affects the main claims and/or conclusions of the paper (regardless of whether the code and data are provided or not)?

Answer: [Yes]

Justification: The paper provides detailed descriptions of all experimental settings, including environment configurations, reward types, input modalities, architecture details, and dataset sizes. We also release all 32 MIKASA-Robo datasets, and implementation details for Online RL, Offline RL, and VLA evaluations are included in the appendices. The benchmark code and data are publicly available at https://tinyurl.com/membenchrobots, ensuring reproducibility of the results and conclusions.

Guidelines:

- The answer NA means that the paper does not include experiments.
- If the paper includes experiments, a No answer to this question will not be perceived well by the reviewers: Making the paper reproducible is important, regardless of whether the code and data are provided or not.
- If the contribution is a dataset and/or model, the authors should describe the steps taken to make their results reproducible or verifiable.
- Depending on the contribution, reproducibility can be accomplished in various ways. For example, if the contribution is a novel architecture, describing the architecture fully might suffice, or if the contribution is a specific model and empirical evaluation, it may be necessary to either make it possible for others to replicate the model with the same dataset, or provide access to the model. In general. releasing code and data is often one good way to accomplish this, but reproducibility can also be provided via detailed instructions for how to replicate the results, access to a hosted model (e.g., in the case of a large language model), releasing of a model checkpoint, or other means that are appropriate to the research performed.

- While NeurIPS does not require releasing code, the conference does require all submissions to provide some reasonable avenue for reproducibility, which may depend on the nature of the contribution. For example
  - (a) If the contribution is primarily a new algorithm, the paper should make it clear how to reproduce that algorithm.
  - (b) If the contribution is primarily a new model architecture, the paper should describe the architecture clearly and fully.
  - (c) If the contribution is a new model (e.g., a large language model), then there should either be a way to access this model for reproducing the results or a way to reproduce the model (e.g., with an open-source dataset or instructions for how to construct the dataset).
  - (d) We recognize that reproducibility may be tricky in some cases, in which case authors are welcome to describe the particular way they provide for reproducibility. In the case of closed-source models, it may be that access to the model is limited in some way (e.g., to registered users), but it should be possible for other researchers to have some path to reproducing or verifying the results.

5. **Open access to data and code**

   Question: Does the paper provide open access to the data and code, with sufficient instructions to faithfully reproduce the main experimental results, as described in supplemental material?

   Answer: [Yes]

   Justification: The paper provides open access to both the benchmark code and all 32 MIKASA-Robo datasets via https://tinyurl.com/membenchrobots. The repository includes detailed instructions for setup, environment configuration, and training, along with scripts for reproducing key experimental results.

   Guidelines:
   - The answer NA means that paper does not include experiments requiring code.
   - Please see the NeurIPS code and data submission guidelines (https://nips.cc/public/guides/CodeSubmissionPolicy) for more details.
   - While we encourage the release of code and data, we understand that this might not be possible, so "No" is an acceptable answer. Papers cannot be rejected simply for not including code, unless this is central to the contribution (e.g., for a new open-source benchmark).
   - The instructions should contain the exact command and environment needed to run to reproduce the results. See the NeurIPS code and data submission guidelines (https://nips.cc/public/guides/CodeSubmissionPolicy) for more details.
   - The authors should provide instructions on data access and preparation, including how to access the raw data, preprocessed data, intermediate data, and generated data, etc.
   - The authors should provide scripts to reproduce all experimental results for the new proposed method and baselines. If only a subset of experiments are reproducible, they should state which ones are omitted from the script and why.
   - At submission time, to preserve anonymity, the authors should release anonymized versions (if applicable).
   - Providing as much information as possible in supplemental material (appended to the paper) is recommended, but including URLs to data and code is permitted.

6. **Experimental setting/details**

   Question: Does the paper specify all the training and test details (e.g., data splits, hyperparameters, how they were chosen, type of optimizer, etc.) necessary to understand the results?

   Answer: [Yes]

   Justification: The paper provides detailed descriptions of training and evaluation settings for all experiments, including the number of trajectories used (e.g., 1000 for Offline RL Subsection 6.3, 250 for VLA Subsection 6.4), observation modalities, action representations, and reward structures. Model architectures, context lengths, chunk sizes, optimizers, and training durations are specified for each baseline.

Guidelines:

- The answer NA means that the paper does not include experiments.
- The experimental setting should be presented in the core of the paper to a level of detail that is necessary to appreciate the results and make sense of them.
- The full details can be provided either with the code, in appendix, or as supplemental material.

7. **Experiment statistical significance**

Question: Does the paper report error bars suitably and correctly defined or other appropriate information about the statistical significance of the experiments?

Answer: [Yes]

Justification: The paper reports mean success rates with standard errors across evaluation episodes for all key experiments, including VLA and Offline RL baselines (e.g., Table 4, Figure 6). Each reported value represents the average over multiple independent rollouts under fixed seeds, capturing variability due to policy stochasticity and environment randomness.

Guidelines:

- The answer NA means that the paper does not include experiments.
- The authors should answer "Yes" if the results are accompanied by error bars, confidence intervals, or statistical significance tests, at least for the experiments that support the main claims of the paper.
- The factors of variability that the error bars are capturing should be clearly stated (for example, train/test split, initialization, random drawing of some parameter, or overall run with given experimental conditions).
- The method for calculating the error bars should be explained (closed form formula, call to a library function, bootstrap, etc.)
- The assumptions made should be given (e.g., Normally distributed errors).
- It should be clear whether the error bar is the standard deviation or the standard error of the mean.
- It is OK to report 1-sigma error bars, but one should state it. The authors should preferably report a 2-sigma error bar than state that they have a 96% CI, if the hypothesis of Normality of errors is not verified.
- For asymmetric distributions, the authors should be careful not to show in tables or figures symmetric error bars that would yield results that are out of range (e.g. negative error rates).
- If error bars are reported in tables or plots, The authors should explain in the text how they were calculated and reference the corresponding figures or tables in the text.

8. **Experiments compute resources**

Question: For each experiment, does the paper provide sufficient information on the computer resources (type of compute workers, memory, time of execution) needed to reproduce the experiments?

Answer: [Yes]

Justification: In the paper and supplementary materials, the authors provide code for training and evaluation, specify random seeds and the number of runs, and detail the compute setup (single NVIDIA A100 GPU with 96 GB RAM), enabling exact reproduction of results (see Appendix G).

Guidelines:

- The answer NA means that the paper does not include experiments.
- The paper should indicate the type of compute workers CPU or GPU, internal cluster, or cloud provider, including relevant memory and storage.
- The paper should provide the amount of compute required for each of the individual experimental runs as well as estimate the total compute.
- The paper should disclose whether the full research project required more compute than the experiments reported in the paper (e.g., preliminary or failed experiments that didn't make it into the paper).

9. **Code of ethics**

   Question: Does the research conducted in the paper conform, in every respect, with the NeurIPS Code of Ethics https://neurips.cc/public/EthicsGuidelines?

   Answer: [Yes]

   Justification: The research adheres to the NeurIPS Code of Ethics. All experiments are conducted in simulation with no human subjects or sensitive data involved. The released benchmark and datasets are open-source and designed to support transparent, reproducible research. The work does not raise concerns related to privacy, fairness, misuse, or environmental impact beyond standard computational practices in the field.

   Guidelines:

   - The answer NA means that the authors have not reviewed the NeurIPS Code of Ethics.
   - If the authors answer No, they should explain the special circumstances that require a deviation from the Code of Ethics.
   - The authors should make sure to preserve anonymity (e.g., if there is a special consideration due to laws or regulations in their jurisdiction).

10. **Broader impacts**

    Question: Does the paper discuss both potential positive societal impacts and negative societal impacts of the work performed?

    Answer: [Yes]

    Justification: The paper discusses broader impacts in the context of advancing memory-intensive RL. On the positive side, the proposed benchmark may accelerate progress toward more capable and reliable autonomous systems in fields such as assistive robotics and household automation. While the work is entirely simulation-based, the authors acknowledge potential concerns regarding the misuse of memory-equipped agents (e.g., in surveillance or manipulation scenarios) and highlight the importance of responsible deployment. These considerations are briefly addressed in the broader impact discussion to guide ethical use.

    Guidelines:

    - The answer NA means that there is no societal impact of the work performed.
    - If the authors answer NA or No, they should explain why their work has no societal impact or why the paper does not address societal impact.
    - Examples of negative societal impacts include potential malicious or unintended uses (e.g., disinformation, generating fake profiles, surveillance), fairness considerations (e.g., deployment of technologies that could make decisions that unfairly impact specific groups), privacy considerations, and security considerations.
    - The conference expects that many papers will be foundational research and not tied to particular applications, let alone deployments. However, if there is a direct path to any negative applications, the authors should point it out. For example, it is legitimate to point out that an improvement in the quality of generative models could be used to generate deepfakes for disinformation. On the other hand, it is not needed to point out that a generic algorithm for optimizing neural networks could enable people to train models that generate Deepfakes faster.
    - The authors should consider possible harms that could arise when the technology is being used as intended and functioning correctly, harms that could arise when the technology is being used as intended but gives incorrect results, and harms following from (intentional or unintentional) misuse of the technology.
    - If there are negative societal impacts, the authors could also discuss possible mitigation strategies (e.g., gated release of models, providing defenses in addition to attacks, mechanisms for monitoring misuse, mechanisms to monitor how a system learns from feedback over time, improving the efficiency and accessibility of ML).

11. **Safeguards**

    Question: Does the paper describe safeguards that have been put in place for responsible release of data or models that have a high risk for misuse (e.g., pretrained language models, image generators, or scraped datasets)?

Answer: [Yes]

Justification: The paper discusses the broader implications of standardizing memory evaluation in RL, which can positively impact the development of more capable and reliable autonomous systems for real-world applications such as assistive robotics and home automation. While the benchmark itself poses minimal direct societal risk, we acknowledge that advances in memory-equipped agents could potentially be misused in surveillance or other sensitive contexts. However, as the work is entirely simulation-based and intended to support open academic research, we believe the positive impacts outweigh the risks. Responsible use and continued community oversight remain essential as memory-centric RL systems mature.

Guidelines:

- The answer NA means that the paper poses no such risks.
- Released models that have a high risk for misuse or dual-use should be released with necessary safeguards to allow for controlled use of the model, for example by requiring that users adhere to usage guidelines or restrictions to access the model or implementing safety filters.
- Datasets that have been scraped from the Internet could pose safety risks. The authors should describe how they avoided releasing unsafe images.
- We recognize that providing effective safeguards is challenging, and many papers do not require this, but we encourage authors to take this into account and make a best faith effort.

12. **Licenses for existing assets**

Question: Are the creators or original owners of assets (e.g., code, data, models), used in the paper, properly credited and are the license and terms of use explicitly mentioned and properly respected?

Answer: [Yes]

Justification: All third-party assets used in this work—including codebases, datasets, and pretrained models—are properly cited with references to their original publications. We use publicly available environments and models (e.g., ManiSkill3, Octo, OpenVLA), each under a permissive open-source license, which we respect in accordance with their terms of use.

Guidelines:

- The answer NA means that the paper does not use existing assets.
- The authors should cite the original paper that produced the code package or dataset.
- The authors should state which version of the asset is used and, if possible, include a URL.
- The name of the license (e.g., CC-BY 4.0) should be included for each asset.
- For scraped data from a particular source (e.g., website), the copyright and terms of service of that source should be provided.
- If assets are released, the license, copyright information, and terms of use in the package should be provided. For popular datasets, paperswithcode.com/datasets has curated licenses for some datasets. Their licensing guide can help determine the license of a dataset.
- For existing datasets that are re-packaged, both the original license and the license of the derived asset (if it has changed) should be provided.
- If this information is not available online, the authors are encouraged to reach out to the asset's creators.

13. **New assets**

Question: Are new assets introduced in the paper well documented and is the documentation provided alongside the assets?

Answer: [Yes]

Justification: The paper introduces two new benchmark suites—MIKASA-Base and MIKASA-Robo—as well as 32 offline RL datasets for robotic memory tasks. All assets are released under a permissive license and are accompanied by comprehensive

documentation, including environment descriptions, task specifications, dataset structure, and usage instructions. The documentation is provided in the code repository (https://tinyurl.com/membenchrobots) to ensure usability and reproducibility by the community.

Guidelines:

- The answer NA means that the paper does not release new assets.
- Researchers should communicate the details of the dataset/code/model as part of their submissions via structured templates. This includes details about training, license, limitations, etc.
- The paper should discuss whether and how consent was obtained from people whose asset is used.
- At submission time, remember to anonymize your assets (if applicable). You can either create an anonymized URL or include an anonymized zip file.

14. **Crowdsourcing and research with human subjects**

Question: For crowdsourcing experiments and research with human subjects, does the paper include the full text of instructions given to participants and screenshots, if applicable, as well as details about compensation (if any)?

Answer: [NA]

Justification: The paper does not involve any crowdsourcing or research with human subjects. All experiments were conducted entirely in simulated environments with no human data collection or interaction.

Guidelines:

- The answer NA means that the paper does not involve crowdsourcing nor research with human subjects.
- Including this information in the supplemental material is fine, but if the main contribution of the paper involves human subjects, then as much detail as possible should be included in the main paper.
- According to the NeurIPS Code of Ethics, workers involved in data collection, curation, or other labor should be paid at least the minimum wage in the country of the data collector.

15. **Institutional review board (IRB) approvals or equivalent for research with human subjects**

Question: Does the paper describe potential risks incurred by study participants, whether such risks were disclosed to the subjects, and whether Institutional Review Board (IRB) approvals (or an equivalent approval/review based on the requirements of your country or institution) were obtained?

Answer: [NA]

Justification: The paper does not involve research with human subjects or any form of data collection from individuals. All experiments were conducted in simulation and do not pose ethical risks requiring IRB or equivalent approval.

Guidelines:

- The answer NA means that the paper does not involve crowdsourcing nor research with human subjects.
- Depending on the country in which research is conducted, IRB approval (or equivalent) may be required for any human subjects research. If you obtained IRB approval, you should clearly state this in the paper.
- We recognize that the procedures for this may vary significantly between institutions and locations, and we expect authors to adhere to the NeurIPS Code of Ethics and the guidelines for their institution.
- For initial submissions, do not include any information that would break anonymity (if applicable), such as the institution conducting the review.

16. **Declaration of LLM usage**

Question: Does the paper describe the usage of LLMs if it is an important, original, or non-standard component of the core methods in this research? Note that if the LLM is used only for writing, editing, or formatting purposes and does not impact the core methodology, scientific rigorousness, or originality of the research, declaration is not required.

Answer: [NA]

Justification: LLMs were only used for spell checking and grammar suggestions.

Guidelines:

- The answer NA means that the core method development in this research does not involve LLMs as any important, original, or non-standard components.
- Please refer to our LLM policy (https://neurips.cc/Conferences/2025/LLM) for what should or should not be described.

