# Table of Contents

## A MIKASA-Robo Implementation Details

An example of running the environment from the MIKASA-Robo benchmark is shown in Code 1. For ease of debugging, we also added various wrappers (found in `mikasa_robo_suite/utils/wrappers/`) that display useful information about the episode on the video (Code 2). Thus, `RenderStepInfoWrapper()` displays the current step in the environment; `DebugRewardWrapper()` displays information about the full reward at the current step in the environment; `DebugRewardWrapper()` displays information about each component that generates the reward function at the current step. In addition, we also added task-specific wrappers for each environment. For example, `RememberColorInfoWrapper()` displays the target color of the cube in the `RememberColor-v0` task, and `ShellGameRenderCupInfoWrapper()` displays which mug the ball is actually under in the `ShellGame-v0` task.

Code 1: Getting started with MIKASA-Robo using the `RememberColor9-v0` environment.

```python
# pip install mikasa_robo_suite
import mikasa_robo_suite
from mikasa_robo_suite.utils.wrappers import
    ↪ StateOnlyTensorToDictWrapper
from tqdm.notebook import tqdm
import torch
import gymnasium as gym

# Create the environment via gym.make()
# obs_mode="rgb" for modes "RGB", "RGB+joint", "RGB+oracle" etc.
# obs_mode="state" for mode "state"
episode_timeout = 90
env = gym.make("RememberColor9-v0", num_envs=512 obs_mode="rgb",
    ↪ render_mode="all")
env = StateOnlyTensorToDictWrapper(env) # * always use this wrapper!

obs, _ = env.reset(seed=42)
print(obs.keys())
for i in tqdm(range(episode_timeout)):
    action = torch.from_numpy(env.action_space.sample())
    obs, reward, terminated, truncated, info = env.step(action)

env.close()
```

Code 2: MIKASA-Robo wrappers system.

```python
import mikasa_robo_suite, torch
from mikasa_robo_suite.dataset_collectors.get_mikasa_robo_datasets
    ↪ import env_info
import gymnasium as gym
from mani_skill.utils.wrappers import RecordEpisode
from IPython.display import Video

env = gym.make("RememberColor9-v0", num_envs=512, obs_mode="rgb",
    ↪ render_mode="all")
state_wrappers_list, episode_timeout = env_info("RememberColor9-v0")
for wrapper_class, wrapper_kwargs in state_wrappers_list:
    env = wrapper_class(env, **wrapper_kwargs)
env = RecordEpisode(env, f"./videos", max_steps_per_video=
    ↪ episode_timeout)

obs, _ = env.reset(seed=42)
for i in range(episode_timeout):
    action = torch.from_numpy(env.action_space.sample())
    obs, reward, terminated, truncated, info = env.step(action)

Video(f"./videos/0.mp4", embed=True, width=640)
env.close()
```

## B  MIKASA-Robo Datasets for Offline RL

To train Offline RL baselines on camera images (in "RGB" mode) with sparse rewards (success condition), we collected datasets for each of the 32 MIKASA-Robo tasks. Datasets were collected using a PPO-MLP agent trained to SR=100% in "state" mode (i.e., with full information about the task being solved) with sparse rewards (success condition). Thus, each dataset is represented by 1000 successful trajectories, where each trajectory consists of:

1. "rgb" (shape: $(T, 128, 128, 6)$) - two RGB images (view from above and from the gripper)
2. "joints" (shape: $(T, 25)$) - Tool Center Point (TCP) position and rotation, and joint positions and velocities
3. "action" (shape: $(T, 8)$) - action (8-dimensional vector)
4. "reward" (shape: $(T, )$) - (dense) reward for each step
5. "success" (shape: $(T,)$) - (sparse) success flag for each step
6. "done" (shape: $(T,)$) - done flag for each step

These datasets are available for download from the project website. We have also published the weights of the PPO-MLP agent used to collect the dataset, as well as scripts for collecting datasets of any size, to our repository.

## C  MIKASA-Base Implementation Details

An example of running an environment from the MIKASA-Base benchmark is shown in Code 3. MIKASA-Base supports the standard Gymnasium API and is fully compatible with all its wrappers. This allows users to leverage various functionalities, including parallelization using `AsyncVectorEnv`. MIKASA-Base provides a predefined set of environments with different levels of difficulty. However, users can customize the environment parameters by passing specific arguments (see Code 3).

Code 3: Example code for running `MemoryLength-v0` environment.

```python
import mikasa_base
import gymnasium as gym

# use pre-defined env
# env_id = "MemoryLengthEasy-v0"
# env_kwargs = None

# create env using custom parameters
env_id = "MemoryLength-v0"
env_kwargs = {"memory_length": 10, "num_bits": 1}
seed = 123

env = gym.make(env_id, env_kwargs)

obs, _ = env.reset(seed=seed)

for i in range(11):
    action = env.action_space.sample()
    next_obs, reward, terminations, truncations, infos = env.step(
        ↪ action)
env.close()
```

## D  MIKASA-Robo setup for VLA baselines

For experiments involving Vision-Language-Action (VLA) models, we focused on a representative subset of spatial and object memory tasks from MIKASA-Robo. For each task, we generated a dataset of 250 episodes using an oracle PPO policy with full access to the environment state. At

Table 5: Tasks configurations for fine-tuning VLA models. The table lists the task ID, number of evaluation steps (T), and the associated language instruction

| Task | T | Language instruction |
|------|---|----------------------|
| RememberColor3/5/9-v0 | 60 | Remember the color of the cube and then pick the matching one |
| ShellGameTouch-v0 | 90 | Memorize the position of the cup covering the ball, then pick that cup |
| InterceptMedium-v0 | 90 | Track the ball's movement, estimate its velocity, then aim the ball at the target |

every timestep, the policy recorded two synchronized RGB frames (one from the static "base" camera and one from the robot's wrist camera) along with the corresponding end-effector control actions ( `pd_ee_delta_pose` controller from [87]). Each task was also paired with a concise language instruction (see Table 5).

All VLA baselines were trained for 50000 iterations and evaluated independently on each task. Complete training/evaluation scripts, language instruction templates, and detailed model hyperparameter settings are provided in the accompanying supplementary code.

## E  Memory Mechanisms in RL

In RL, memory mechanisms are techniques or models used to enable agents to retain and recall information from past interactions with the environment.

There are several approaches to incorporating memory into RL, including recurrent neural networks (RNNs) [76, 37, 14] which uses hidden states to store information from previous steps [93, 34], state-space models (SSMs) [28, 83, 27] which uses system state to store historical information [31, 77], transformers [92] which uses attention mechanism to capture sequential dependencies inside the context window [72, 54, 68], graph neural networks (GNNs) [98] which uses graphs to store information [99, 44] etc. Popular agents with memory mechanisms are summarized in Table 2.

## F  Classic baselines performance on the MIKASA-Robo benchmark

In this section, we present a comprehensive evaluation of PPO-MLP and PPO-LSTM baselines on our MIKASA-Robo benchmark. Our experiments with PPO-MLP in `state` mode using dense rewards demonstrate perfect performance across all tasks, consistently achieving 100% success rate, as shown in Figure 7 and Figure 8. This remarkable performance serves as a crucial validation of our benchmark design: when an agent has access to complete state information and receives dense rewards, it can master these tasks completely. Therefore, any performance degradation in `RGB+joints` mode observed with other algorithms or training configurations must stem from the algorithmic limitations or learning challenges rather than any inherent flaws in the task design. This empirical evidence confirms that our environments are well-calibrated and properly designed, establishing a solid foundation for evaluating memory-enhanced algorithms. All results are presented as mean $\pm$ standard error of the mean (SEM), where the mean is computed across three independent training runs, and each trained agent is evaluated on 16 different random seeds to ensure robust performance assessment.

The performance evaluation of PPO-MLP and PPO-LSTM with dense rewards in the `RGB+joints` mode is presented in Figure 9. This mode specifically tests the agents' memory capabilities, as it requires remembering and utilizing historical information to solve the tasks. Our results demonstrate a clear distinction between memory-less and memory-enhanced architectures, while also revealing the limitations of conventional memory mechanisms.

Consider the `RememberColor-v0` environment as an illustrative example. In its simplest configuration with three cubes, the memory-less PPO-MLP achieves only 25% success rate. In contrast, PPO-LSTM, leveraging its memory mechanism, achieves perfect performance with 100% success rate. However, as task complexity increases to five or nine cubes, even the LSTM's memory capabilities prove insufficient, with performance degrading significantly.

These results validate two key aspects of our benchmark: first, its effectiveness in distinguishing between memory-less and memory-enhanced architectures, and second, its ability to challenge even sophisticated memory mechanisms as task complexity increases. This demonstrates that

MIKASA-Robo provides a competitive yet meaningful evaluation framework for developing and testing advanced memory-enhanced agents.

Our evaluation of PPO-MLP and PPO-LSTM baselines under sparse reward conditions in `RGB+joints` mode reveals the true challenge of our benchmark tasks. As shown in Figure 10, both architectures – even the memory-enhanced LSTM – consistently fail to achieve any meaningful success rate across nearly all considered environments. This striking result underscores the extreme difficulty of memory-intensive manipulation tasks when only terminal rewards are available, highlighting the substantial gap between current algorithms and the level of memory capabilities required for real-world robotic applications.

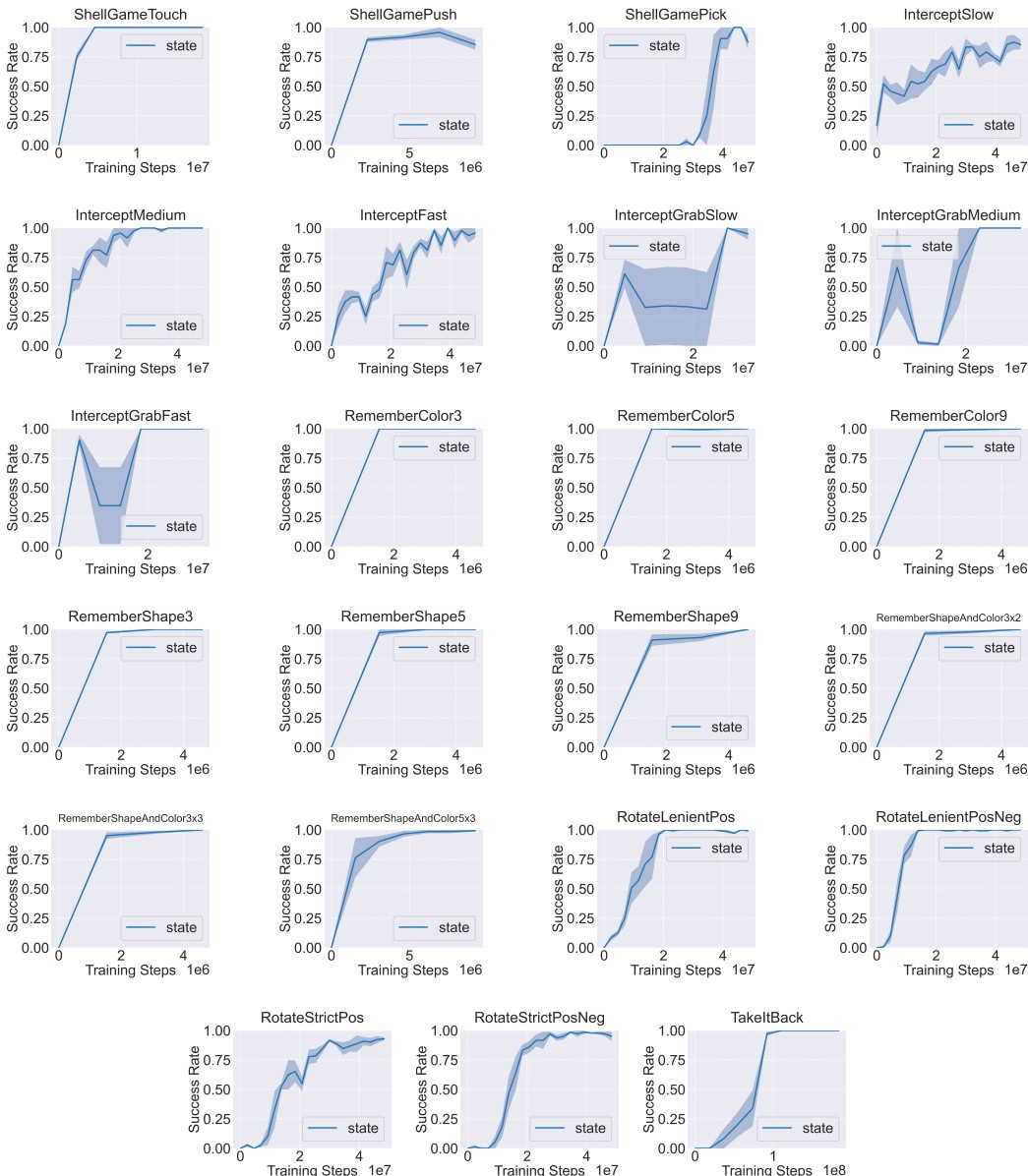

Figure 7: Demonstration of PPO-MLP performance on MIKASA-Robo benchmark when trained with oracle-level `state` information. In this learning mode, MDP problem formulation is considered, i.e. memory is not required for successful problem solving. At the same time, the obtained results show that it is possible to solve these problems and obtain 100% Success Rate.

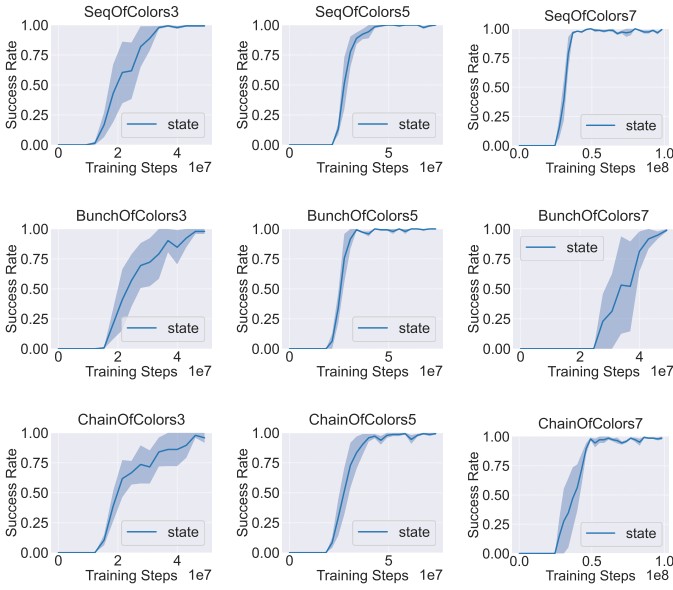

Figure 8: Demonstration of PPO-MLP performance on MIKASA-Robo benchmark when trained with oracle-level `state` information. Results are shown for memory capacity (`SeqOfColors[3,5,7]-v0`, `BunchOfColors[3,5,7]-v0`) and sequential memory (`ChainOfColors[3,5,7]-v0`).

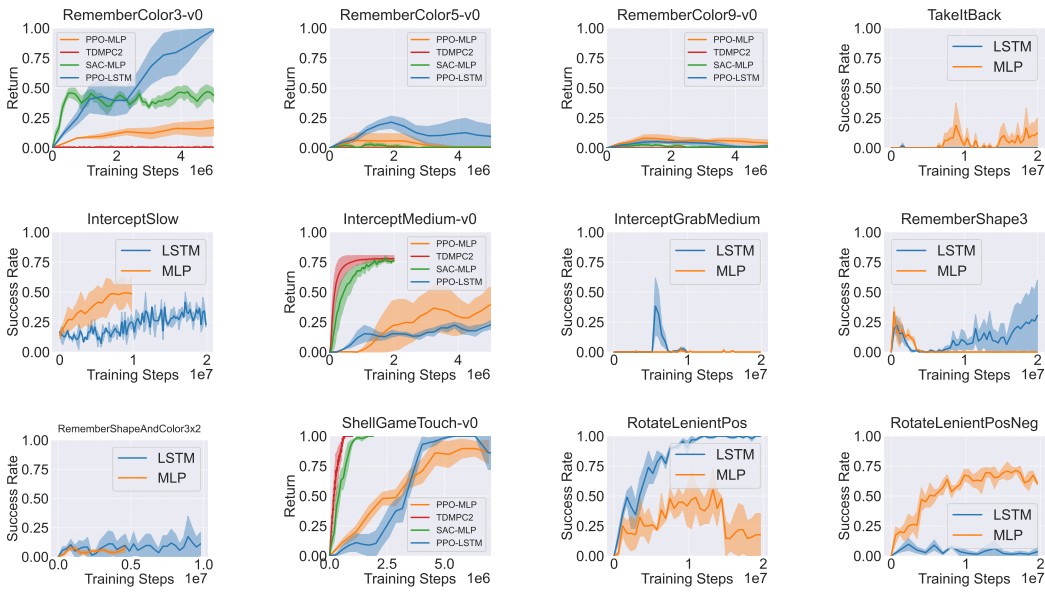

Figure 9: Performance evaluation of PPO-MLP and PPO-LSTM on the MIKASA-Robo benchmark using the "RGB+joints" training mode with dense reward function, where the agent only receives images from the camera (from above and from the gripper) and information about the state of the joints (position and velocity). The results demonstrate that numerous tasks pose significant challenges even for PPO-LSTM agents with memory, establishing these environments as effective benchmarks for evaluating advanced memory-enhanced architectures.

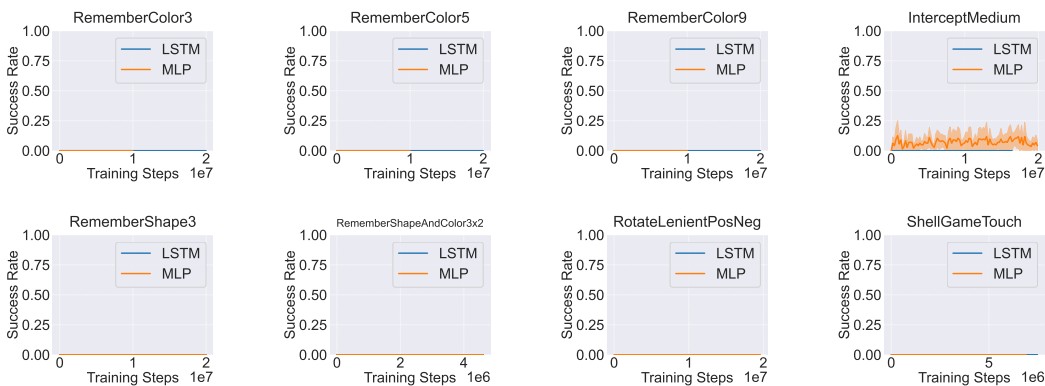

Figure 10: Performance evaluation of PPO-MLP and PPO-LSTM on the MIKASA-Robo benchmark using the "RGB+joints" with sparse reward function training mode, where the agent only receives images from the camera (from above and from the gripper) and information about the state of the joints (position and velocity). This training mode with sparse reward function causes even more difficulty for the agent to learn, making this mode even more challenging for memory-enhanced agents.

# G   Experiments Reproducing and Compute Resources

All baselines were trained and evaluated under a reproducible standardized setup. Training for every algorithm was performed on a single NVIDIA A100 GPU. For evaluation, each task was run for 100 independent episodes, with environment and agent random seeds ranging from 1 to 100. We report performance metrics as the mean success rate $\pm$ the standard error of the mean (SEM) over these 100 trials.

# H   MIKASA-Robo Detailed Tasks Description

In this section, we provide comprehensive descriptions of the 32 memory-intensive tasks that comprise the MIKASA-Robo benchmark. Each task is designed to evaluate specific aspects of memory capabilities in robotic manipulation, ranging from object tracking and spatial memory to sequential decision-making. For each task, we detail its objective, memory requirements, observation space, reward structure, and success criteria. Additionally, we explain how task complexity increases across different variants and discuss the specific memory challenges they present. The following subsections describe each task category and its variants in detail.

Each of the proposed environment supports multiple observation modes:

- **State**: Full state information including ball position
- **RGB+joints**: Two camera views (top-down and gripper) plus robot joint states
- **RGB**: Only visual information from two cameras

In the case of `RotateLenient-v0` and `RotateStrict-v0`, the prompt information available at each step is additionally added to each observation.

Table 6: **Results for Offline RL baselines**. The table shows comparison of transformer-based baselines (RATE, DT), behavior cloning (BC), classic Offline RL baselines (CQL), and Diffusion Policy (DP) on all 32 tasks from the MIKASA-Robo benchmark. Results are presented as mean ± sem across the three runs, where each run is averaged over 100 episodes and sem is the standard error of the mean. Training was performed using only RGB observations (two cameras: top view and gripper view) and using sparse rewards (success once condition). The results show that even models with memory (RATE, DT) are not able to solve most of the benchmark problems, which makes it challenging and promising for further validation of the algorithm.

| # | Environment | RATE | DT | BC | CQL | DP |
|---|---|---|---|---|---|---|
| 1 | ShellGameTouch-v0 | 0.92±0.01 | 0.53±0.07 | 0.28±0.01 | 0.16±0.04 | 0.18±0.02 |
| 2 | ShellGamePush-v0 | 0.78±0.06 | 0.62±0.14 | 0.27±0.01 | 0.25±0.01 | 0.22±0.03 |
| 3 | ShellGamePick-v0 | 0.02±0.01 | 0.00±0.00 | 0.01±0.01 | 0.00±0.00 | 0.01±0.00 |
| 4 | InterceptSlow-v0 | 0.23±0.02 | 0.40±0.02 | 0.37±0.06 | 0.25±0.01 | 0.33±0.05 |
| 5 | InterceptMedium-v0 | 0.32±0.02 | 0.56±0.01 | 0.31±0.14 | 0.03±0.01 | 0.68±0.02 |
| 6 | InterceptFast-v0 | 0.30±0.04 | 0.36±0.04 | 0.03±0.02 | 0.02±0.02 | 0.21±0.05 |
| 7 | InterceptGrabSlow-v0 | 0.09±0.03 | 0.00±0.00 | 0.28±0.18 | 0.03±0.00 | 0.03±0.01 |
| 8 | InterceptGrabMedium-v0 | 0.09±0.03 | 0.00±0.00 | 0.11±0.02 | 0.08±0.04 | 0.03±0.01 |
| 9 | InterceptGrabFast-v0 | 0.14±0.03 | 0.11±0.03 | 0.09±0.02 | 0.08±0.03 | 0.18±0.02 |
| 10 | RotateLenientPos-v0 | 0.11±0.04 | 0.01±0.01 | 0.15±0.03 | 0.16±0.02 | 0.11±0.02 |
| 11 | RotateLenientPosNeg-v0 | 0.29±0.03 | 0.05±0.02 | 0.22±0.01 | 0.12±0.02 | 0.14±0.05 |
| 12 | RotateStrictPos-v0 | 0.03±0.02 | 0.05±0.04 | 0.01±0.00 | 0.03±0.01 | 0.06±0.02 |
| 13 | RotateStrictPosNeg-v0 | 0.08±0.01 | 0.05±0.03 | 0.04±0.02 | 0.04±0.02 | 0.15±0.01 |
| 14 | TakeItBack-v0 | 0.42±0.24 | 0.08±0.04 | 0.33±0.10 | 0.04±0.01 | 0.05±0.02 |
| 15 | RememberColor3-v0 | 0.65±0.04 | 0.01±0.01 | 0.27±0.03 | 0.29±0.01 | 0.32±0.01 |
| 16 | RememberColor5-v0 | 0.13±0.03 | 0.07±0.05 | 0.12±0.01 | 0.15±0.02 | 0.10±0.02 |
| 17 | RememberColor9-v0 | 0.09±0.02 | 0.01±0.01 | 0.12±0.02 | 0.15±0.01 | 0.17±0.01 |
| 18 | RememberShape3-v0 | 0.21±0.04 | 0.05±0.04 | 0.31±0.04 | 0.20±0.10 | 0.32±0.05 |
| 19 | RememberShape5-v0 | 0.17±0.04 | 0.04±0.04 | 0.18±0.01 | 0.15±0.00 | 0.21±0.04 |
| 20 | RememberShape9-v0 | 0.05±0.00 | 0.05±0.02 | 0.10±0.02 | 0.14±0.01 | 0.11±0.02 |
| 21 | RememberShapeAndColor3x2-v0 | 0.14±0.02 | 0.04±0.02 | 0.13±0.02 | 0.11±0.05 | 0.14±0.02 |
| 22 | RememberShapeAndColor3x3-v0 | 0.08±0.03 | 0.06±0.06 | 0.09±0.02 | 0.09±0.02 | 0.16±0.01 |
| 23 | RememberShapeAndColor5x3-v0 | 0.07±0.02 | 0.01±0.01 | 0.09±0.01 | 0.09±0.02 | 0.11±0.03 |
| 24 | BunchOfColors3-v0 | 0.00±0.00 | 0.00±0.00 | 0.00±0.00 | 0.00±0.00 | 0.00±0.00 |
| 25 | BunchOfColors5-v0 | 0.00±0.00 | 0.00±0.00 | 0.00±0.00 | 0.00±0.00 | 0.00±0.00 |
| 26 | BunchOfColors7-v0 | 0.00±0.00 | 0.00±0.00 | 0.00±0.00 | 0.00±0.00 | 0.00±0.00 |
| 27 | SeqOfColors3-v0 | 0.00±0.00 | 0.00±0.00 | 0.00±0.00 | 0.00±0.00 | 0.00±0.00 |
| 28 | SeqOfColors5-v0 | 0.00±0.00 | 0.00±0.00 | 0.00±0.00 | 0.00±0.00 | 0.00±0.00 |
| 29 | SeqOfColors7-v0 | 0.00±0.00 | 0.00±0.00 | 0.00±0.00 | 0.00±0.00 | 0.00±0.00 |
| 30 | ChainOfColors3-v0 | 0.00±0.00 | 0.00±0.00 | 0.00±0.00 | 0.00±0.00 | 0.00±0.00 |
| 31 | ChainOfColors5-v0 | 0.00±0.00 | 0.00±0.00 | 0.00±0.00 | 0.00±0.00 | 0.00±0.00 |
| 32 | ChainOfColors7-v0 | 0.00±0.00 | 0.00±0.00 | 0.00±0.00 | 0.00±0.00 | 0.00±0.00 |

1354

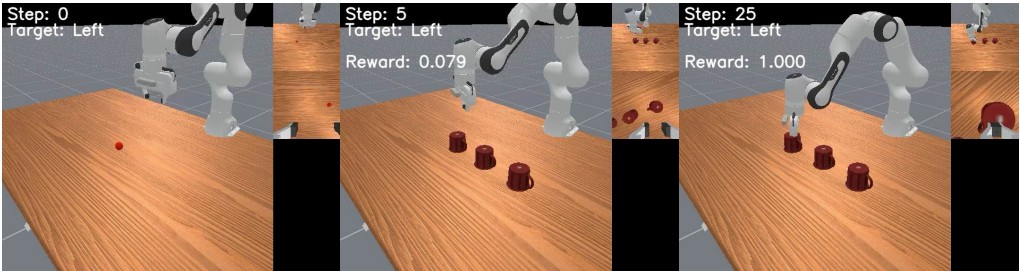

Figure 11: `ShellGameTouch-v0`: The robot observes a ball in front of it. next, this ball is covered by a mug and then the robot has to touch the mug with the ball underneath.

## H.1 ShellGame-v0

The `ShellGame-v0` task (Figure 11) is inspired by a simplified version of the classic shell game, which tests a person's ability to remember object locations when they become occluded. This task evaluates an agent's capacity for object permanence and spatial memory, crucial skills for real-world robotic manipulation where objects frequently become temporarily hidden from view.

**Environment Description**   The environment consists of three identical mugs placed on a table and a red ball. The task proceeds in three phases:

1. **Observation Phase** (steps 0-4): The ball is placed at one of three positions, and the agent can observe its location.

2. **Occlusion Phase** (step 5): The ball and positions are covered by three identical mugs.

3. **Action Phase** (steps 6+): The agent must interact with the mug covering the ball's location. The type of target interaction depends on the selected mode: `Touch`, `Push` and `Pick`.

**Task Modes**   The task includes three variants of increasing difficulty:

- `Touch`: The agent only needs to touch the correct mug
- `Push`: The agent must push the correct mug to a designated area
- `Pick`: The agent must pick and lift the correct mug above a specified height

**Success Criteria**   Success is determined by:

- `Touch`: Contact between the gripper and the correct mug
- `Push`: Moving forward the correct mug to the target zone
- `Pick`: Elevating the correct mug above 0.1m

**Reward Structure**   The environment provides both sparse and dense reward variants:

- **Sparse**: Binary reward (1.0 for success, 0.0 otherwise)
- **Dense**: Continuous reward based on:
  - Distance between gripper and target mug
  - Robot's motion smoothness (static reward based on joint velocities)
  - Task completion status (additional reward when the task is solved)

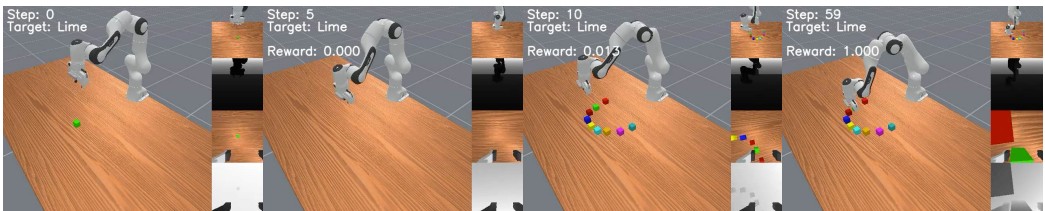

Figure 12: `RememberColor9-v0`: The robot observes a colored cube in front of it, then this cube disappears and an empty table is shown. Then 9 cubes appear on the table, and the agent must touch a cube of the same color as the one it observed at the beginning of the episode.

## H.2 RememberColor-v0

The `RememberColor-v0` task (Figure 12) tests an agent's ability to remember and identify objects based on their visual properties. This capability is essential for real-world robotics applications where agents must recall and match object characteristics across time intervals.

**Environment Description** The environment presents a sequence of colored cubes on a table. The task proceeds in three phases:

1. **Observation Phase** (steps 0-4): A cube of a specific color is displayed, and the agent must memorize its color.
2. **Delay Phase** (steps 5-9): The cube disappears, leaving an empty table.
3. **Selection Phase** (steps 10+): Multiple cubes of different colors appear (3, 5, or 9 depending on difficulty), and the agent must identify and interact with the cube matching the original color.

**Task Modes** The task includes three complexity levels:

- 3 (easy): Choose from 3 different colors (red, lime, blue)
- 5 (Medium): Choose from 5 different colors (red, lime, blue, yellow, magenta)
- 9 (Hard): Choose from 9 different colors (red, lime, blue, yellow, magenta, cyan, maroon, olive, teal)

**Success Criteria** Success is determined by:

- Correctly identifying and touching the cube that matches the color shown in the observation phase
- Maintaining contact with the correct cube for at least 0.1 seconds

**Reward Structure** The environment provides both sparse and dense reward variants:

- **Sparse**: Binary reward (1.0 for success, 0.0 otherwise)
- **Dense**: Continuous reward based on:
  - Distance between gripper and target cube
  - Robot's motion smoothness (static reward based on joint velocities)
  - Additional reward for robot being static while touching the correct cube
  - Task completion status (additional reward when the task is solved)

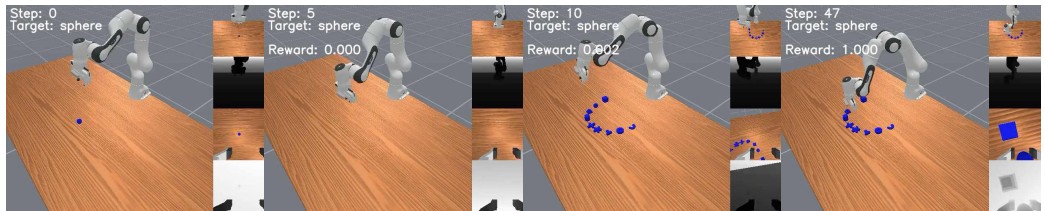

Figure 13: `RememberShape9-v0`: The robot observes an object with specific shape in front of it, then the object disappears and an empty table appears. Then 9 objects of different shapes appear on the table, and the agent must touch an object of the same shape as the one it observed at the beginning of the episode.

### H.3 RememberShape-v0

The `RememberShape-v0` task (Figure 13) evaluates an agent's ability to remember and identify objects based on their geometric properties. This capability is crucial for robotic applications where shape recognition and recall are essential for successful manipulation.

**Environment Description** The environment presents a sequence of geometric shapes on a table. The task proceeds in three phases:

1. **Observation Phase** (steps 0-4): A shape (cube, sphere, cylinder, etc.) is displayed, and the agent must memorize its geometry.
2. **Delay Phase** (steps 5-9): The shape disappears, leaving an empty table.
3. **Selection Phase** (steps 10+): Multiple shapes appear (3, 5, or 9 depending on difficulty), and the agent must identify and interact with the shape matching the original geometry.

**Task Modes** The task includes three complexity levels:

- 3 (Easy): Choose from 3 different shapes (cube, sphere, cylinder)
- 5 (Medium): Choose from 5 different shapes (cube, sphere, cylinder cross, torus)
- 9 (Hard): Choose from 9 different shapes (cube, sphere, cylinder cross, torus, star, pyramid, t-shape, crescent)

**Success Criteria** Success is determined by:

- Correctly identifying and touching the object with the same shape shown in the observation phase
- Maintaining contact with the correct shape for at least 0.1 seconds

**Reward Structure** The environment provides both sparse and dense reward variants:

- **Sparse**: Binary reward (1.0 for success, 0.0 otherwise)
- **Dense**: Continuous reward based on:
    - Distance between gripper and target object
    - Robot's motion smoothness (static reward based on joint velocities)
    - Additional reward for maintaining static position when touching correct object
    - Task completion status (additional reward when the task is solved)

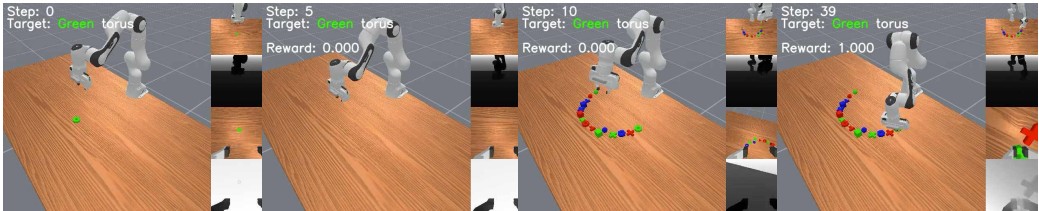

Figure 14: `RememberShapeAndColor5x3-v0`: An object of a certain shape and color appears in front of the agent. Then the object disappears and the agent sees an empty table. Then objects of 5 different shapes and 3 different colors appear on the table and the agent has to touch what it observed at the beginning of the episode.

## H.4 RememberShapeAndColor-v0

The `RememberShapeAndColor-v0` task (Figure 14) evaluates an agent's ability to remember and identify objects based on multiple visual properties simultaneously. This task combines shape and color recognition, testing the agent's capacity to maintain and match multiple object features across time intervals.

**Environment Description** The environment presents a sequence of colored geometric shapes on a table. The task proceeds in three phases:

1. **Observation Phase** (steps 0-4): An object with specific shape and color is displayed, and the agent must memorize both properties.

2. **Delay Phase** (steps 5-9): The object disappears, leaving an empty table.

3. **Selection Phase** (steps 10+): Multiple objects with different combinations of shapes and colors appear, and the agent must identify and interact with the object matching both the original shape and color.

**Task Modes** The task includes three complexity levels based on the number of shape-color combinations:

- `3x2` (Easy): Choose from 6 objects (3 shapes × 2 colors); shapes: cube, sphere, t-shape; colors: red, green

- `3x3` (Medium): Choose from 9 objects (3 shapes × 3 colors); shapes: cube, sphere, t-shape; colors: red, green, blue

- `5x3` (Hard): Choose from 15 objects (5 shapes × 3 colors); shapes: cube, sphere, t-shape, cross, torus; colors: red, green, blue

**Success Criteria** Success is determined by:

- Correctly identifying and touching the object that matches both the shape and color shown in the observation phase

- Maintaining contact with the correct object for at least 0.1 seconds

**Reward Structure** The environment provides both sparse and dense reward variants:

- **Sparse**: Binary reward (1.0 for success, 0.0 otherwise)

- **Dense**: Continuous reward based on:
  - Distance between gripper and target object
  - Robot's motion smoothness (static reward based on joint velocities)
  - Additional reward for maintaining static position while touching correct object
  - Task completion status (additional reward when the task is solved)

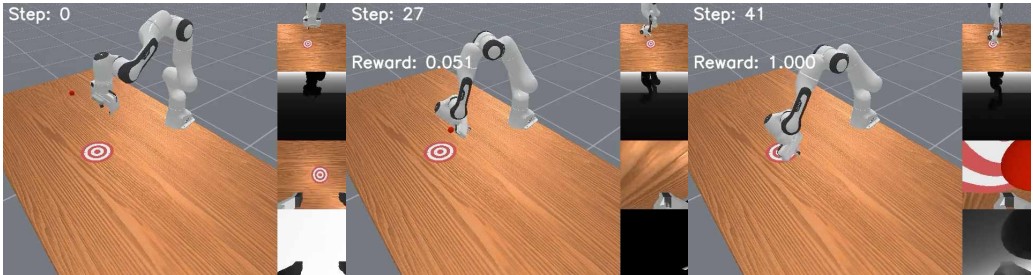

Figure 15: `InterceptMedium-v0`: A ball rolls on the table in front of the agent with a random initial velocity, and the agent's task is to intercept this ball and direct it at the target zone.

## H.5  Intercept-v0

The `Intercept-v0` task (Figure 16) evaluates an agent's ability to predict and intercept a moving object based on its initial trajectory. This task tests the agent's capacity for motion prediction and spatial-temporal reasoning, which are essential skills for dynamic manipulation tasks in robotics.

**Environment Description**  The environment consists of a red ball moving across a table and a target zone. The task requires the agent to:

1. Observe the ball's initial position and velocity
2. Predict the ball's trajectory
3. Guide the ball to reach a designated target zone

**Task Modes**  The task includes three variants with increasing ball velocities:

- `Slow`: Ball velocity range of 0.25-0.5 m/s
- `Medium`: Ball velocity range of 0.5-0.75 m/s
- `Fast`: Ball velocity range of 0.75-1.0 m/s

**Success Criteria**  Success is determined by:

- Guiding the ball to enter the target zone
- The ball must come to rest within the target area (radius 0.1m)

**Reward Structure**  The environment provides both sparse and dense reward variants:

- **Sparse**: Binary reward (1.0 for success, 0.0 otherwise)
- **Dense**: Continuous reward based on:
    - Distance between gripper and ball
    - Distance between ball and target zone
    - Robot's motion smoothness (static reward based on joint velocities)
    - Task completion status (additional reward when the task is solved)

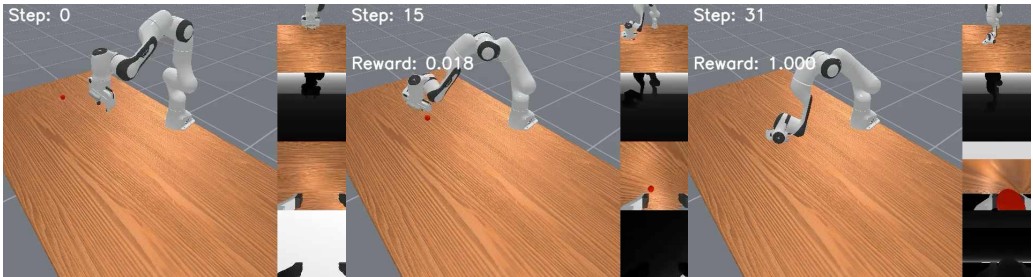

Figure 16: `InterceptGrabMedium-v0`: A ball rolls on the table in front of the agent with a random initial velocity, and the agent's task is to intercept this ball with a gripper and lift it up.

## H.6   InterceptGrab-v0

The `InterceptGrab-v0` task (Figure 16) extends the `Intercept-v0` task by requiring the agent to not only predict the trajectory of a moving object but also grasp it while in motion. This task evaluates the agent's ability to combine motion prediction with precise manipulation timing, simulating real-world scenarios where robots must catch or intercept moving objects.

**Environment Description**   The environment consists of a red ball moving across a table. The task requires the agent to:

1. Observe the ball's initial position and velocity
2. Predict the ball's trajectory
3. Position the gripper to intercept the ball's path
4. Time the grasping action correctly to catch the ball
5. Maintain a stable grasp while bringing the ball to rest

**Task Modes**   The task includes three variants with increasing ball velocities:

- `Slow`: Ball velocity range of 0.25-0.5 m/s
- `Medium`: Ball velocity range of 0.5-0.75 m/s
- `Fast`: Ball velocity range of 0.75-1.0 m/s

**Success Criteria**   Success is determined by:

- Successfully grasping the moving ball
- Maintaining a stable grasp until the ball comes to rest
- The robot must be static with the ball firmly grasped

**Reward Structure**   The environment provides both sparse and dense reward variants:

- **Sparse**: Binary reward (1.0 for success, 0.0 otherwise)
- **Dense**: Continuous reward based on:
  - Distance between gripper and ball
  - Grasping reward
  - Robot's motion smoothness (static reward based on joint velocities)
  - Task completion status (additional reward when the task is solved)

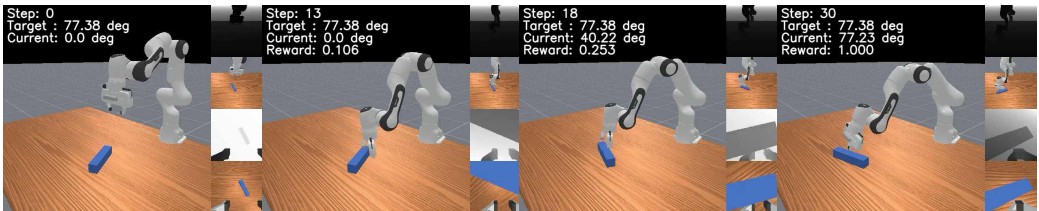

Figure 17: `RotateLenientPos-v0`: A randomly oriented peg is placed in front of the agent. The agent's task is to rotate this peg by a certain angle (the center of the peg can be shifted).

### H.7 RotateLenient-v0

The `RotateLenient-v0` task (Figure 17) evaluates an agent's ability to remember and execute specific rotational movements. This task tests the agent's capacity to maintain and reproduce angular information, which is crucial for manipulation tasks requiring precise orientation control. This task tests the agent's ability to hold information in memory about how far peg has already rotated at the current step relative to its initial position.

**Environment Description** The environment consists of a blue-colored peg on a table that must be rotated by a specified angle. The task proceeds in one phase, but the static prompt information about the target angle is available to the agent at each timestep:

    1. **Action Phase**: The agent must rotate the peg to match the target angle

**Task Modes** The task includes two variants with different rotation requirements:

- `Pos`: Rotate by a positive angle between 0 and $\pi/2$
- `PosNeg`: Rotate by either positive or negative angle between $-\pi/4$ and $\pi/4$

**Success Criteria** Success is determined by:

- Rotating the peg to within the angle threshold (±0.1 radians) of the target angle
- Maintaining the final orientation in a stable position
- The robot must be static with the peg at the correct orientation

**Reward Structure** The environment provides both sparse and dense reward variants:

- **Sparse**: Binary reward (1.0 for success, 0.0 otherwise)
- **Dense**: Continuous reward based on:
  - Distance between gripper and peg
  - Angular distance to target rotation
  - Stability of the peg's orientation
  - Robot's motion smoothness (static reward based on joint velocities)
  - Task completion status (additional reward when the task is solved)

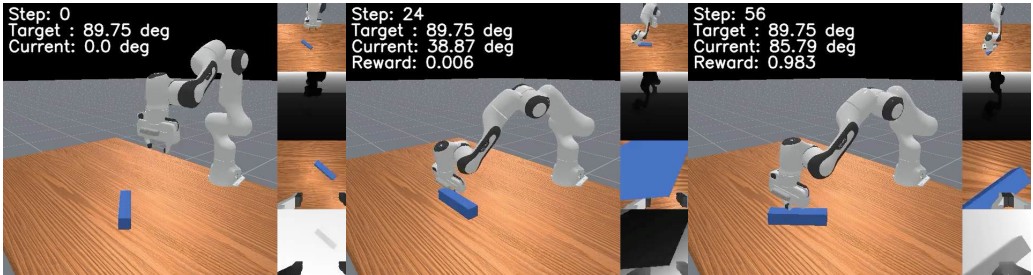

Figure 18: `RotateStrictPos-v0`: A randomly oriented peg is placed in front of the agent. The agent's task is to rotate this peg by a certain angle (it is not allowed to move the center of the peg)

## H.8 RotateStrict-v0

The `RotateStrict-v0` task (Figure 18) extends the `RotateLenient-v0` task with more stringent requirements for precise rotational control.

**Environment Description** The environment consists of a blue-colored peg on a table that must be rotated by a specified angle while maintaining its position. The task proceeds in one phase, but the static prompt information about the target angle is available to the agent at each timestep:

1. **Action Phase**: The agent must rotate the peg to match the target angle while keeping it centered

**Task Modes** The task includes two variants with different rotation requirements:

- `Pos`: Rotate by a positive angle between 0 and $\pi/2$
- `PosNeg`: Rotate by either positive or negative angle between $-\pi/4$ and $\pi/4$

**Success Criteria** Success is determined by:

- Rotating the peg to within the angle threshold (±0.1 radians) of the target angle
- Maintaining the peg's position within 5cm of its initial XY coordinates
- The robot must be static with the peg at the correct orientation
- No significant deviation in other rotation axes

**Reward Structure** The environment provides both sparse and dense reward variants:

- **Sparse**: Binary reward (1.0 for success, 0.0 otherwise)
- **Dense**: Continuous reward based on:
  - Distance between gripper and peg
  - Angular distance to target rotation
  - Position deviation from initial location
  - Stability of the peg's orientation
  - Robot's motion smoothness (static reward based on joint velocities)
  - Task completion status (additional reward when the task is solved)

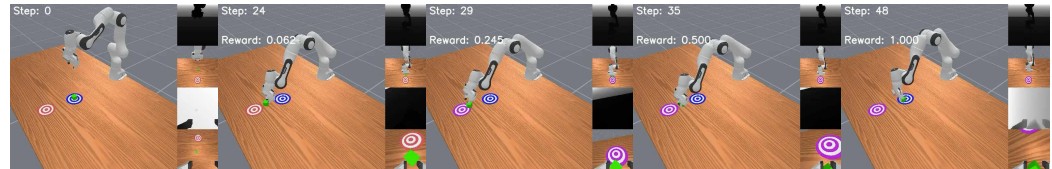

Figure 19: `TakeItBack-v0`: The agent observes a green cube in front of him. The agent's task is to move the green cube to the red target, and as soon as it lights up violet, return the cube to its original position (the agent does not observes the original position of the cube).

## H.9    TakeItBack-v0

The `TakeItBack-v0` task (Figure 19) assesses the agent's ability to perform sequential tasks and memorize the starting position. This task tests the agent's capacity for sequential memory and spatial reasoning, requiring it to maintain information about past locations and achievements while executing a multi-step plan.

**Environment Description**    The environment consists of a green cube and two target regions (initial and goal) on a table. The task proceeds in two phases:

1. **First Phase**: The agent must move the cube from its initial position to a goal region
2. **Second Phase**: After reaching the goal, goal region change it's color from red to magenta, and the agent must return the cube to its original position (marked by the initial region and invisible for the agent)

**Success Criteria**    Success is determined by:

- First reaching the goal region with the cube
- Then returning the cube to the initial region
- Both goals must be achieved in sequence

**Reward Structure**    The environment provides both sparse and dense reward variants:

- **Sparse**: Binary reward (1.0 for success, 0.0 otherwise)
- **Dense**: Continuous reward based on:
  - Distance between gripper and cube
  - Distance to current target region
  - Progress through the task sequence
  - Stability of cube manipulation
  - Robot's motion smoothness (static reward based on joint velocities)
  - Task completion status (additional reward when the task is solved)

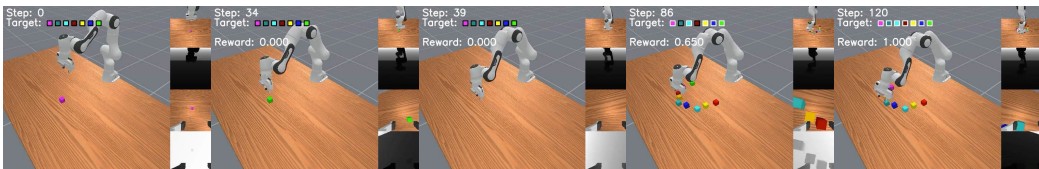

Figure 20: `SeqOfColors7-v0`: In front of the agent, 7 cubes of different colors appear sequentially. After the last cube is shown, the agent observes an empty table. Then 9 cubes of different colors appear on the table and the agent has to touch the cubes that were shown at the beginning of the episode in any order.

## H.10 SeqOfColors-v0

The `SeqOfColors-v0` task (Figure 20) evaluates an agent's ability to remember and reproduce an unordered sequence of colors. This task tests memory capacity capabilities essential for robotic tasks that require following specific patterns or sequences.

**Environment Description**   The environment presents a sequence of colored cubes that must be reproduced in any order. The task proceeds in two phases:

1. **Observation Phase** (steps $0$-$(5N - 1)$): A sequence of N colored cubes is shown one at a time, with each cube visible for 5 steps.
2. **Delay Phase** (steps $(5N)$-$(5N + 4)$): All cubes disappear
3. **Selection Phase** (steps $(5N + 5)$+): A larger set of cubes appears, and the agent must identify and touch all previously shown cubes in any order

**Task Modes**   The task includes three complexity levels:

- 3 (Easy): Remember 3 colors demonstrated sequentially
- 5 (Medium): Remember 5 colors demonstrated sequentially
- 7 (Hard): Remember 7 colors demonstrated sequentially

**Success Criteria**   Success is determined by:

- Correctly identifying and touching all cubes from the observation phase
- Order of selection doesn't matter
- Each cube must be touched for at least 0.1 seconds
- The demonstrated set must be touched without any mistakes

**Reward Structure**   The environment provides both sparse and dense reward variants:

- **Sparse**: Binary reward (1.0 for success, 0.0 otherwise)
- **Dense**: Continuous reward based on:
  - Distance between gripper and next target cube
  - Number of correctly identified cubes
  - Static reward for stable contact
  - Robot's motion smoothness (static reward based on joint velocities)
  - Task completion status (additional reward when the task is solved)

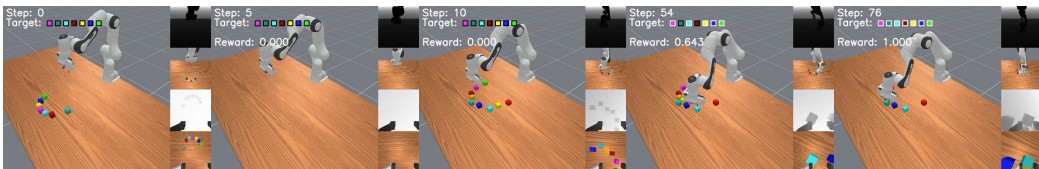

Figure 21: `BunchOfColors7-v0`: 7 cubes of different colors appear simultaneously in front of the agent. After the agent observes an empty table. Then, 9 cubes of different colors appear on the table and the agent has to touch the cubes that were shown at the beginning of the episode in any order.

## H.11  BunchOfColors-v0

The `BunchOfColors-v0` task (Figure 21) tests an agent's memory capacity by requiring it to remember multiple objects simultaneously. This capability is crucial for tasks requiring parallel processing of multiple items.

**Environment Description**    The environment presents multiple colored cubes simultaneously. The task proceeds in three phases:

1. **Observation Phase** (steps 0-4): Multiple colored cubes are displayed simultaneously
2. **Delay Phase** (steps 5-9): All cubes disappear
3. **Selection Phase** (steps 10+): A larger set of cubes appears, and the agent must identify and touch all previously shown cubes in any order

**Task Modes**    The task includes three complexity levels:

- 3 (Easy): Remember 3 colors demonstrated simultaneously
- 5 (Medium): Remember 5 colors demonstrated simultaneously
- 7 (Hard): Remember 7 colors demonstrated simultaneously

**Success Criteria**    Success is determined by:

- Correctly identifying and touching all cubes from the observation phase
- Order of selection doesn't matter
- Each cube must be touched for at least 0.1 seconds
- The demonstrated set must be touched without any mistakes

**Reward Structure**    The environment provides both sparse and dense reward variants:

- **Sparse**: Binary reward (1.0 for success, 0.0 otherwise)
- **Dense**: Continuous reward based on:
  - Distance between gripper and next target cube
  - Static reward for stable contact
  - Number of correctly touched cubes
  - Robot's motion smoothness (static reward based on joint velocities)
  - Task completion status (additional reward when the task is solved)

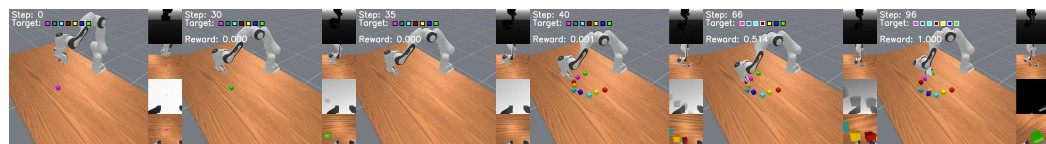

Figure 22: `ChainOfColors7-v0`: In front of the agent, 7 cubes of different colors appear sequentially. After the last cube is shown, the agent sees an empty table. Then 9 cubes of different colors appear on the table and the agent must unmistakably touch the cubes that were shown at the beginning of the episode, in the same strict order.

## H.12 ChainOfColors-v0

The `ChainOfColors-v0` task (Figure 22) evaluates the agent's ability to store and retrieve ordered information. This task simulates scenarios where the agent must track changing relationships between objects over time.

**Environment Description** The environment presents am ordered sequence (chain) of colored cubes that must be followed. The task proceeds in multiple phases:

1. **Observation Phase** (steps $0$-$(5N-1)$): A sequence of N colored cubes is shown one at a time, with each cube visible for 5 steps.
2. **Delay Phase** (steps $(5N)$-$(5N+4)$): All cubes disappear
3. **Selection Phase** (steps $(5N+5)$+): A larger set of cubes appears, and the agent must identify and touch all previously shown cubes in the exact order as demonstrated

**Task Modes** The task includes three complexity levels:

- 3 (Easy): Remember 3 colors demonstrated sequentially
- 5 (Medium): Remember 5 colors demonstrated sequentially
- 7 (Hard): Remember 7 colors demonstrated sequentially

**Success Criteria** Success is determined by:

- Correctly identifying and touching all cubes from the observation phase in the exact order
- Each cube must be touched for at least 0.1 seconds
- The demonstrated set must be touched without any mistakes

**Reward Structure** The environment provides both sparse and dense reward variants:

- **Sparse**: Binary reward (1.0 for success, 0.0 otherwise)
- **Dense**: Continuous reward based on:
  - Distance between gripper and next target cube
  - Static reward for stable contact
  - Number of correctly touched cubes
  - Robot's motion smoothness (static reward based on joint velocities)
  - Task completion status (additional reward when the task is solved)

Table 7: Classification of environments from the MIKASA-Base benchmark according to the suggested memory-intensive tasks classification from the Subsection 4.2.

| Environment | Memory Task | Brief description of the task | Observation Space | Action Space |
|---|---|---|---|---|
| Memory Cards | Capacity | Memorize the positions of revealed cards and correctly match pairs while minimizing incorrect guesses. | vector | discrete |
| Numpad | Sequential | Memorize the sequence of movements and navigate the rolling ball on a 3×3 grid by following the correct order while avoiding mistakes. | image, vector | discrete, continuous |
| BSuite Memory Length | Object | Memorize the initial context signal and recall it after a given number of steps to take the correct action. | vector | discrete |
| Minigrid-Memory | Object | Memorize the object in the starting room and use this information to select the correct path at the junction. | image | discrete |
| Ballet | Sequential, Object | Memorize the sequence of movements performed by each uniquely colored and shaped dancer, then identify and approach the dancer who executed the given pattern. | image | discrete |
| Passive Visual Match | Object | Memorize the target color displayed on the wall during the initial phase. After a brief distractor phase, identify and select the target color among the distractors by stepping on the corresponding ground pad. | image | discrete |
| Passive-T-Maze | Object | Memorize the goal's location upon initial observation, navigate through the maze with limited sensory input, and select the correct path at the junction. | vector | discrete |
| ViZDoom-two-colors | Object | Memorize the color of the briefly appearing pillar (green or red) and collect items of the same color to survive in the acid-filled room. | image | discrete |
| Memory Maze | Spatial | Memorize the locations of objects and the maze structure using visual clues, then navigate efficiently to find objects of a specific color and score points. | image | discrete |
| MemoryGym Mortar Mayhem | Capacity, Sequential | Memorize a sequence of movement commands and execute them in the correct order. | image | discrete |
| MemoryGym Mystery Path | Capacity, Spatial | Memorize the invisible path and navigate it without stepping off. | image | discrete |
| POPGym Repeat First | Object | Memorize the initial value presented at the first step and recall it correctly after receiving a sequence of random values. | vector | discrete |
| POPGym Repeat Previous | Sequential, Object | Memorize the value observed at each step and recall the value from $k$ steps earlier when required. | vector | discrete |
| POPGym Autoencode | Sequential | Memorize the sequence of cards presented at the beginning and reproduce them in the same order when required. | vector | discrete |
| POPGym Count Recall | Object, Capacity | Memorize unique values encountered and count how many times a specific value has appeared. | vector | discrete |
| POPGym vectorless Cartpole | Sequential | Memorize velocity data over time and integrate it to infer the position of the pole for balance control. | vector | continuous |
| POPGym vectorless Pendulum | Sequential | Memorize angular velocity over time and integrate it to infer the pendulum's position for successful swing-up control. | vector | continuous |
| POPGym Multiarmed Bandit | Object, Capacity | Memorize the reward probabilities of different slot machines by exploring them and identify the one with the highest expected reward. | vector | discrete |
| POPGym Concentration | Capacity | Memorize the positions of revealed cards and match them with previously seen cards to find all matching pairs. | vector | discrete |
| POPGym Battleship | Spatial | Memorize the coordinates of previous shots and their HIT or MISS feedback to build an internal representation of the board, avoid repeat shots, and strategically target ships for maximum rewards. | vector | discrete |
| POPGym Mine Sweeper | Spatial | Memorize revealed grid information and use numerical clues to infer safe tiles while avoiding mines. | vector | discrete |
| POPGym Labyrinth Explore | Spatial | Memorize previously visited cells and navigate the maze efficiently to discover new, unexplored areas and maximize rewards. | vector | discrete |
| POPGym Labyrinth Escape | Spatial | Memorize the maze layout while exploring and navigate efficiently to find the exit and receive a reward. | vector | discrete |
| POPGym Higher Lower | Object, Sequential | Memorize previously revealed card ranks and predict whether the next card will be higher or lower, updating the reference card after each prediction to maximize rewards. | vector | discrete |

# I MIKASA-Base Benchmark Tasks Description

This section provides a detailed description of all environments included in the MIKASA-Base benchmark Section 5. Understanding the characteristics and challenges of these environments is crucial for evaluating RL algorithms. Each environment presents unique tasks, offering diverse scenarios to test the memory abilities of RL agents.

## I.1 Memory Cards

The Memory Cards environment [19] is a memory game environment with 5 randomly shuffled pairs of hidden cards. At each step, the agent sees one revealed card and must find its matching pair. A correct guess removes both cards; otherwise, the card is hidden again, and a new one is revealed. The game continues until all pairs are removed.

## I.2 Numpad

The Numpad environment [38] consists of an $N \times N$ grid of tiles. The agent controls a ball that rolls between tiles. At the beginning of an episode, a random sequence of $n$ neighboring tiles (excluding diagonals) is selected, and the agent must follow this sequence in the correct order. The environment is structured so that pressing the correct tile lights it up, while pressing an incorrect tile resets progress. A reward of +1 is given for the first press of each correct tile after a reset. The episode ends after a fixed number of steps. To succeed, the agent must memorize the sequence and navigate it correctly without mistakes. The ability to "jump" over tiles is not available.

### I.3 BSuite MemoryLength

The MemoryLength environment [70] represents a sequence of observations, where at each step, the observation takes a value of either +1 or -1. The environment is structured so that a reward is given only at the final step if the agent correctly predicts the $i$-th value from the initial observation vector $obs$. The index of this $i$-th value is specified at the last step observation vector in $obs[1]$. To succeed, the agent must remember the sequence of observations and use this information to make an accurate prediction at the final step.

### I.4 Minigrid-Memory

Minigrid-Memory [12] is a two-dimensional grid-based environment that features a T-shaped maze with a small room at the beginning of the corridor, containing an object. The agent starts at a random position within the corridor. Its task is to reach the room, observe and memorize the object, then proceed to the junction at the maze's end and turn towards the direction where an identical object is located. The reward function is defined as $R_t = 1 - 0.9 \times \frac{t}{T}$ for a successful attempt; otherwise, the agent receives zero reward. The episode terminates when the agent makes a choice at the junction or exceeds a time limit of steps.

### I.5 Ballet

In the Ballet environment [54] tasks take place in an $11 \times 11$ tiled room, consisting of a $9 \times 9$ central area surrounded by a one-tile-wide wall. Each tile is upsampled to 9 pixels, resulting in a $99 \times 99$ pixel input image. The agent is initially placed at the center of the room, while dancers are randomly positioned in one of 8 possible locations around it. Each dancer has a distinct shape and color, selected from 15 possible shapes and 19 colors, ensuring uniqueness. These visual features serve only for identification and do not influence behavior. The agent itself is always represented as a white square. The agent receives egocentric visual observations, meaning its view is centered on its own position, which has been shown to enhance generalization.

### I.6 Passive T-Maze

The Passive T-Maze environment [68] consists of a corridor leading to a junction that connects two possible goal states. The agent starts at a designated position and can move in four directions: left, right, up, or down. At the beginning of each episode, one of the two goal states is randomly assigned as the correct destination. The agent observes this goal location before starting its movement. The agent stays in place if it attempts to move into a wall. To succeed, the agent must navigate to the correct goal based on its initial observation. The optimal strategy involves moving through the corridor towards the junction and then selecting the correct path.

### I.7 ViZDoom-Two-Colors

The ViZDoom-Two-Colors [84] is an environment where an agent is placed in a room with constantly depleting health. The room contains red and green objects, one of which restores health (+1 reward), while the other reduces it (-1 reward). The beneficial color is randomly assigned at the beginning of each episode and indicated by a column. The environment is structured so that the agent must memorize the column's color to collect the correct items. Initially, the column remains visible, but in a harder variant, it disappears after 45 steps, increasing the memory requirement. To succeed, the agent must maximize survival by collecting beneficial objects while avoiding harmful ones.

### I.8 Memory Maze

The Memory Maze environment [73] is a procedurally generated 3D maze. Each episode, the agent spawns in a new maze with multiple colored objects placed in fixed locations. The agent receives a first-person view and a prompt indicating the color of the target object. It must navigate the maze, memorize object positions, and return to them efficiently. The agent receives a reward of 1 for reaching the correct object and no reward for incorrect objects.

### I.9 MemoryGym Mortar Mayhem

Mortar Mayhem [75] is a grid-based environment where the agent must memorize and execute a sequence of commands in the correct order. The environment consists of a finite grid, where the agent initially observes a series of movement instructions and then attempts to reproduce them accurately. Commands include movements to adjacent tiles or remaining in place. The challenge lies in the agent's ability to recall and execute a growing sequence of instructions, with failure resulting in episode termination. A reward of +0.1 is given for each correctly executed command

### I.10 MemoryGym Mystery Path

Mystery Path [75] presents an invisible pathway that the agent must traverse without deviating. If the agent steps off the path, it is returned to the starting position, forcing it to remember the correct trajectory. The path is procedurally generated, meaning each episode introduces a new configuration. Success in this environment requires the agent to accurately recall previous steps and missteps to avoid repeating errors. The agent is rewarded +0.1 for progressing onto a previously unvisited tile

### I.11 POPGym environments

The following environments are included from the POPGym benchmark [65], which is designed to evaluate RL agents in partially observable settings. POPGym provides a diverse collection of lightweight vectorized environments with varying difficulty levels.

#### I.11.1 POPGym Autoencode

The environment consists of a deck of cards that is shuffled and sequentially shown to the agent during the watch phase. While observing the cards, a watch indicator is active, but it disappears when the last card is revealed. Afterward, the agent must reproduce the sequence of cards in the correct order. The environment is structured to evaluate the agent's ability to encode a sequence of observations into an internal representation and later reconstruct the sequence one observation at a time.

#### I.11.2 POPGym Concentration

The environment represents a classic memory game where a shuffled deck of cards is placed face-down. The agent sequentially flips two cards and earns a reward if the revealed cards form a matching pair. The environment is designed in such a way that the agent must remember previously revealed cards to maximize its success rate.

#### I.11.3 POPGym Repeat First

The environment presents the agent with an initial value from a set of four possible values, along with an indicator signaling that this is the first value. In subsequent steps, the agent continues to receive random values from the same set but without the initial indicator. The structure requires the agent to retain the first received value in memory and recall it accurately to receive a reward.

#### I.11.4 POPGym Repeat Previous

The environment consists of a sequence of observations, where each observation can take one of four possible values at each timestep. The agent is tasked with recalling and outputting the value that appeared a specified number of steps in the past.

#### I.11.5 POPGym Stateless Cartpole

This is a modified version of the traditional Cartpole environment [6] where angular and linear position information is removed from observations. Instead, the agent only receives velocity-based data and must infer positional states by integrating this information over time to successfully balance the pole.

### I.11.6 POPGym Stateless Pendulum

In this variation of the swing-up pendulum environment [18], angular position data is omitted from the agent's observations. The agent must infer the pendulum's position by processing velocity information and use this estimate to determine appropriate control actions.

### I.11.7 POPGym Noisy Stateless Cartpole

This environment builds upon Stateless Cartpole by introducing Gaussian noise into the observations. The agent must still infer positional states from velocity information while filtering out the added noise to maintain control of the pole.

### I.11.8 POPGym Noisy Stateless Pendulum

This variation extends the Stateless Pendulum environment by incorporating Gaussian noise into the observations. The agent must manage this uncertainty while using velocity data to estimate the pendulum's position and swing it up effectively.

### I.11.9 POPGym Multiarmed Bandit

The Multiarmed Bandit environment is an episodic formulation of the multiarmed bandit problem [82], where a set of bandits is randomly initialized at the start of each episode. Unlike conventional multiarmed bandit tasks, where reward probabilities remain fixed across episodes, this structure resets them every time. The agent must dynamically adjust its exploration and exploitation strategies to maximize long-term rewards.

### I.11.10 POPGym Higher Lower

Inspired by the higher-lower card game, this environment presents the agent with a sequence of cards. At each step, the agent must predict whether the next card will have a higher or lower rank than the current one. Upon making a guess, the next card is revealed and becomes the new reference. The agent can enhance its performance by employing card counting strategies to estimate the probability of future values.

### I.11.11 POPGym Count Recall

At each timestep, the agent is presented with two values: a next value and a query value. The agent must determine and output how many times the query value has appeared so far. To succeed, the agent must maintain an accurate count of past occurrences and retrieve the correct number upon request.

### I.11.12 POPGym Battleship

A partially observable variation of the game Battleship, where the agent does not have access to the full board. Instead, it receives feedback on its previous shot, indicating whether it was a HIT or MISS, along with the shot's location. The agent earns rewards for hitting ships, receives no reward for missing, and incurs a penalty for targeting the same location more than once. The environment challenges the agent to construct an internal representation of the board and update its strategy based on past observations.

### I.11.13 POPGym Mine Sweeper

A partially observable version of the computer game Mine Sweeper, where the agent lacks direct visibility of the board. Observations include the coordinates of the most recently clicked tile and the number of adjacent mines. Clicking on a mined tile results in a negative reward and ends the game. To succeed, the agent must track previous selections and deduce mine locations based on the numerical clues, ensuring it avoids mines while uncovering safe tiles.

### I.11.14 POPGym Labyrinth Explore

The environment consists of a procedurally generated 2D maze in which the agent earns rewards for reaching new, unexplored tiles. Observations are limited to adjacent tiles, requiring the agent to infer the larger maze layout through exploration. A small penalty per timestep incentivizes efficient navigation and discovery strategies.

### I.11.15 POPGym Labyrinth Escape

This variation of Labyrinth Explore challenges the agent to find an exit rather than merely exploring the maze. The agent retains the same restricted observation space, seeing only nearby tiles. Rewards are only given upon successfully reaching the exit, making it a sparse reward environment where the agent must navigate strategically to achieve its goal.