# OpenReview forum: "Memory, Benchmark & Robots: A Benchmark for Solving Complex Tasks with Reinforcement Learning"
_NeurIPS.cc/2025/Datasets_and_Benchmarks_Track — Submitted to NeurIPS 2025 Datasets and Benchmarks Track_

### Official Review · Reviewer_XNMz · 2025-06-30

**Rating:** 5
**Confidence:** 5

**Summary:**

The paper presents MIKASA, a benchmark focused on evaluating memory in reinforcement-learning agents. It offers a four-category taxonomy (object, spatial, sequential, capacity memory) and two components: MIKASA-Base, a Gym-style collection of existing memory tasks, and MIKASA-Robo, 32 new tabletop-manipulation environments that systematically vary memory demands. Baseline tests with RL and sequence-modeling algorithms.

**Additional Feedback:**

The questions can be found in the Weaknesses.

**Dataset Code Accessibility:**

Yes

**Dataset Code Comments:**

The authors have provided the code to the benchmark repository, which supports the claim of releasing code and data.

**Ethical Comments:**

Its applications are confined to simulation environments, and no direct societal risks or ethical issues are apparent.

**Ethical Considerations:**

No, there are no or only very minor ethics concerns

**Final Justification:**

I have read the authors’ rebuttal carefully and appreciate their detailed and constructive response. The clarifications are helpful and address the primary concerns raised. While some minor limitations remain, such as aspects of distractor robustness, the rebuttal strengthens the submission overall. I see no critical flaws in the rebuttal, and I maintain my original score. I continue to lean towards acceptance.

**Limitations Weaknesses:**

**Weaknesses (Stong):**

**1. Long Horizon Scenarios:** The benchmark tasks, while well-designed for memory, appear constrained to relatively short temporal horizons and modest memory complexity. For instance, the *RememberColor* task only requires recalling an attribute among a handful of options after a brief delay. The paper does not cover long-range dependencies or sequential tasks, which is a key hallmark of human-level memory.

**2. Claims:** The paper states that “MIKASA bridges the gap between human-like memory complexity and RL-agent requirements.” This assertion feels overstated given the benchmark’s current scope. True “human-like” memory involves far more than episodic recall within current simulation: it encompasses very long-horizon recall, continual accumulation of experiences across multiple sessions, transfer and reuse of memories across distinct tasks and domains, hierarchical storage of semantic, episodic, and procedural knowledge, selective forgetting, and robust retrieval under noisy or ambiguous cues etc. The present benchmark, although valuable in its own way, does not comprise of these elements. Thus, the claim of bridging the gap to human-like memory should be omitted.

**Weaknesses (Minor):**

**3.  Generalization and Robustness Evaluation:** How do the authors assess generalization in this benchmark, and what evidence is there that agents aren’t overfitting to the specifics of the tasks? I am interested in whether the evaluation goes beyond the training distribution. I am not sure if the model memorizes the training scenarios (ironic as that might be in a memory benchmark). In the robotic tasks, do the authors use randomization (different object positions, colors, distractor objects) to ensure an agent must generalize its memory strategy? For example, in the ShellGameTouch, what if the ball colour is changed at the test time? The question also extends to robustness against distractors or irrelevant observations: did the authors measure performance when additional non-essential elements are present? If not, would they expect a big drop if, say, extra objects or misleading cues are introduced? Any insight into how unseen scenarios affect performance (or plans to include such tests in future) would be appreciated.

**Strengths Contributions:**

**Strengths:**

**1. Clear Motivation and Importance:** Memory is a critical but underexplored aspect of embodied AI. The paper rightly emphasizes that many tasks require long-term memory, yet most existing benchmarks (e.g. ALFRED, CALVIN etc.) focus on short-term planning or language grounding without explicitly testing memory. By focusing squarely on memory, the paper fills a genuine gap in the benchmarking landscape.

**2. Good Writing:** The paper is clear, concise, and well-organized. Its explanations of task setup, motivation, and evaluation approach are logically structured, facilitating straightforward comprehension of the methodology and results. Definitions of memory types and task specifications are clear. Overall, the submission is very well-written and effectively communicates its key ideas and experiments.

**3. Well-Designed and Diverse Tasks:** The proposed tasks explicitly challenge the agent’s memory. This is analogous to specialized memory benchmarks like Memory Maze or MemoryGym, but extended to realistic 3D environments. Also, the tasks are also diverse, strengthening the benchmark.

**4. Good range of baselines:** The authors evaluate online RL, offline RL, model-based methods, and VLA architectures, offering practitioners a useful performance spectrum.

---

> ### Author Rebuttal · Authors · 2025-07-31
>
> We thank you for your time spent on our work, as well as for your useful and constructive comments on our work. We also thank you for noting the diversity of our benchmark tasks and the wide range of baselines.
>
> **1. Long Horizon Scenarios**
>
> Our primary objective is to demonstrate whether an agent must use memory at all, rather than to study long horizon scenarios. In many tasks (for instance, RememberColor) the moment the target is hidden the process becomes non-Markovian, so a 60-step episode already exposes memory shortfalls.
>
> Researchers who wish to explore truly long horizons can freely adjust every memory-related setting. All environments ship with convenient knobs for episode length and timing. For example, in RememberColor we can stretch the episode from 60 to 1 000 steps (can be set to any arbitrarily large number), keep the cube visible for 50 steps instead of 5, and extend the occlusion interval from 5 to 900 steps; the code snippet below shows how to do this in some lines of code:
>
> ```python
> from mani_skill.utils.registration import register_env
> from mikasa_robo_suite.remember_color import RememberColorBaseEnv
> import gymnasium as gym
>
> @register_env("RememberColor3Debug-v0", max_episode_steps=1000)
> class RememberColorDebugEnv(RememberColorBaseEnv):
> 	COLORS    	= 3 	# 1-9 unique cubes
> 	TIME_OFFSET   = 50   # how long target cube is shown
> 	GOAL_THRESH   = 0.03  # goal threshold
> 	CUBE_HALFSIZE = 0.02  # cubes size
> 	DELTA_TIME	= 900   # empty table duration
>
> env = gym.make("RememberColor3Debug-v0", num_envs=256, obs_mode="rgb", render_mode="all", delta_time=DELTA_TIME)
> ```
>
> Other tasks are customizable in a similar way, we will add extended instructions to the camera-ready version.
>
> “The paper does not cover sequential tasks, which is a key hallmark of human-level memory.” - In our paper, we introduce a separate type of sequential memory, which is the focus of the proposed BunchOfColors, SeqOfColors, and ChainOfColors tasks. These tasks can also be customizable by the user in a few lines of code. For example, the task SeqOfColors:
>
> ```python
> from mani_skill.utils.registration import register_env
> from mikasa_robo_suite.seq_of_colors import SeqOfColorsEnv
> import gymnasium as gym
>
> @register_env("SeqOfColors6Debug-v0", max_episode_steps=1000)
> class RememberShapeAndColorDebugEnv(SeqOfColorsEnv):
> 	COLORS      	= 2 	# 1-9 unique cubes
> 	GOAL_THRESH 	= 0.03  # goal threshold
> 	CUBE_HALFSIZE   = 0.02  # cubes size
> 	SEQUENCE_LENGTH = 2 	# Number of cubes to show in sequence (1-9)
> 	STEP_DURATION   = 15	# Duration to show each cube
> 	EMPTY_DURATION  = 5 	# Duration of empty table
>
> env = gym.make("SeqOfColors6Debug-v0", num_envs=256, obs_mode="rgb", render_mode="all")
> ```
>
> **2. Claims**
>
> Our intention was not to suggest that MIKASA fully evaluates or reproduces human‐level memory (Figure 3); rather, we wanted to emphasise that it sits between two extremes: 1) Very simple RL memory tests (e.g. frame stacking in Atari or mazes), which miss many real-robot challenges, and 2) True human-like memory (long-term, semantic, episodic, procedural, with selective forgetting, etc.) which is far beyond the scope of today’s simulators.
>
> What MIKASA does aim to “bridge” is the conceptual gap: it translates well-studied cognitive notions (object permanence, spatial recall, sequence memory, capacity limits) into manipulation tasks that are still tractable for current.  To better reflect this intent and avoid potential misunderstandings, we will revise the phrasing in Figure 3 and the accompanying text to more clearly communicate the scope of our benchmark. Thank you for pointing this out. we appreciate your careful reading.
>
> **3. Generalization and Robustness Evaluation**
>
> Our benchmark uses two levels of randomization: scene-level and memory-level.
> Thus, at the scene level, noise is added to the positions of all objects (balls, cubes, spheres, crosses, targets, etc.) during initialization, random rotation of peg in the RotateLenient and RotateStrict tasks, and random velocity of ball in the Intercept and InterceptGrab tasks. A small amount of noise is also applied to the robot joints.
> In turn, the memory-level uses randomization of parameters affecting the memory check. For example, in ShellGame tasks one of three positions of ball to appear is randomly selected; in RememberColor tasks the color of the target cube and other cubes is randomly selected from the specified maximum number of colors, and also the order in which all cubes are placed on the table after time delay is randomly selected; in RememberShape tasks the same is done, but with shapes of objects (9 types of shapes are available); in SeqOfColors tasks the sequence of colors of cubes is randomly selected, and then after time delay the order in which cubes are placed on the table is randomly selected, etc.
>
> Thus, in each of the tasks, randomization occurs at two levels: scene-level and memory-level, guaranteeing high generalizability without overtraining.
>
> We would also like to note that during validation we used a set of random seeds that does not occur during training, i.e. validation distribution != training distribution.
>
> Directly in ShellGame the color of ball is fixed and always red, but, for example, in the RememberShape task the user is allowed to choose an arbitrary color:
>
> ```python
> from mani_skill.utils.registration import register_env
> from mikasa_robo_suite.remember_shape import RememberShapeBaseEnv
> import gymnasium as gym
>
> @register_env("RememberShape6Debug-v0", max_episode_steps=1000)
> class RememberShapeDebugEnv(RememberShapeBaseEnv):
> 	SHAPES    	= 6 	# 1-9 unique shapes
> 	TIME_OFFSET   = 200   # how long target cube is shown
> 	GOAL_THRESH   = 0.03  # more difficult goal threshold
> 	SHAPE_SCALE   = 0.02  # cubes size
> 	COLOR = [0, 0, 255, 255] # each object has the same color
> 	DELTA_TIME	= 100   # empty table duration (seconds)
>
> env = gym.make("RememberShape6Debug-v0", num_envs=256, obs_mode="rgb", render_mode="all", delta_time=DELTA_TIME)
> ```
>
> In future benchmark updates, we plan to extend this functionality to allow explicit control over the selection and partitioning of object attributes such as colors and shapes.
>
> As for distractors or irrelevant observations - in ShellGame tasks, starting from the moment when all three positions (including the one on which the ball is located) are hidden by cups, extra cups present in the scene can be perceived as distractors. Moreover, in the RememberColor, RememberShape, RememberShapeAndColor tasks, the robot has to select only one target object from the 9 or 15 located on the table (see Figure 14 in Appendix, H.4), while in the BunchOfColors, SeqOfColors, and ChainOfColors tasks, the agent has to select only a subsequence of cubes from the whole set (9 cubes). Thus, distractors or irrelevant observations are present in these tasks.
>
> While cluttered scenes primarily challenge perception and motion planning rather than memory itself, we agree that they can provide a useful source of variability and realism. In future benchmark updates, we plan to extend each task with an optional cluttered variant, where a varying number of distractor objects are placed in the scene, either simple geometric shapes (as in Figure 14) or more realistic items from the YCB dataset.
>
>
> We again thank the reviewer for his constructive and helpful comments and hope that our responses have clarified the understanding of our work.

---

> > ### Comment · Reviewer_XNMz · 2025-08-05
> >
> > Thanks for the detailed rebuttal. The authors have addressed my concerns in a satisfactory manner, and I have no further questions at this stage. I see no critical flaws in the current version, and the paper remains a solid and well-motivated contribution to its area. I maintain my score and continue to lean toward acceptance, as I believe the benchmark has strong potential for adoption and impact within the community.

---

> > > ### Author Response · Authors · 2025-08-06
> > >
> > > We appreciate your positive feedback on our work. We are very happy that you found our paper a solid and well-motivated contribution to the area.

---

### Official Review · Reviewer_7Tpx · 2025-07-01

**Rating:** 3
**Confidence:** 4

**Summary:**

This work contributes a new robotic manipulation benchmark focused on evaluating memory-intensive tasks. The authors suggest that existing benchmarks on memory do not effectively test the memory demands of robotic manipulation tasks. Their new benchmark involves a robotic simulator and a set of new tasks designed to span a taxonomy of memory task types they propose. They also perform some baselines on this task, using online and offline classical RL algorithms, and an initial study of VLMs on the task.

**Dataset Code Accessibility:**

Yes

**Dataset Code Comments:**

Looks good!

**Ethical Considerations:**

No, there are no or only very minor ethics concerns

**Final Justification:**

My main issue with this paper is that the tasks feel like "toy tasks that happen to involve a robot arm" vs realistic manipulation tasks involving memory. And the tasks still feel very simple/trivial - the fact that existing algorithms fail on them speaks to the limitations of existing algorithms, but we should be designing benchmarks to be ready for future, more capable algorithms.

**Limitations Weaknesses:**

- not sure sec 3.3 adds much
- the paper doesn't clearly explain why there'a separate "base" and "robo" environment. "base" is poorly explained and justified - the whole intro of the paper is explaining why we need a better robotics-specific memory env, not a better general memory benchmark. and key figures/explanation is missing - the figures on https://sites.google.com/view/memorybenchrobots/ are helpful but aren't even in the paper/supplement? this feels like an afterthought; either remove it or tie it in better with the intro and the rest of the paper. i think it should be removed though since it's not used in any of the experiments either.
- table 6 referenced but not in main paper, and supp table 6 is not relevant?
- does intercept task really require memory? seems like there would be strategies that don't really maybe
- overall the tasks seem quite simple/trivial. i was hoping for some more complex and realistic things that also required memory. like for the cups one, you could do the classic "shuffle the cups for a while and then require it to pick the cup with the ball under it". i was also hoping for some with much longer horizons - the longest is 180 timesteps? eg i think a modern large VLM (coupled to some kind of action policy) would find all of these trivial, since they could all probably fit in a context window. (i still think this despite your later VLM experiments). more clutter could be another area of added complexity.
- the tasks also feel pretty "synthetic" - eg they're mostly like the classic memory puzzles, but now you have to control a robot arm to choose the answer. this feels kinda lame; wasn't the point of this to have tasks that are more like real challenges in robotic manipulation? like, a more practical and difficult task could be to show the filling of a box with various items (hiding the target item), and then request it to pick out the target from under the clutter.
- fig 6 legend is confusing. category labels should not be here.
- there should be an analog of fig 6 for the online algorithms too?
- different options for observation space are nice.
- would be nice to present the results of offline algs on non-sparse rewards too?
- i think the VLA/VLM direction is the right way to go with solving these types of tasks, and i appreciate the use of expert trajectories to bootstrap the model vs just starting with random exploration, which isn't really productive in sparse tasks. but i'm not super convinced these experiments were performed very thoroughly, which you acknowledge yourself.


If the confusing "MIKASA-Base" thing were cleaned up/removed, and some harder/more realistic tasks added, I would feel good about moving this to an accept. The baselines don't feel very strong but the contribution of this type of work is primarily in the benchmark IMO.

**Strengths Contributions:**

- i generally agree with the claim that existing memory tasks don't translate super well to robotic manipulation tasks, and thus a more targeted benchmark could be useful here.
- i appreciate the work done to make a taxonomy of task types and ensure coverage of each. this is helpful for evaluating the general strengths and weaknesses of different algs
- using a modern robotic simulator is nice
- pretty well written and clear.

---

> ### Author Rebuttal · Authors · 2025-07-31
>
> We thank the reviewer for the detailed feedback and for recognising the need for a memory-focused manipulation benchmark.
>
> Below we respond to each concern in the same order.
>
> **1. Section 3.3.**
>
> The paragraph was added to emphasise that MIKASA-Robo targets tabletop manipulation exclusively - not navigation or mobile manipulation.
>
> **2. Separate “base” and “robo” environment**
>
> We have added “base” environments to our benchmark as a set of diagnostic tasks under a single convenient API for initial quick and easy verification of memory mechanisms' performance before their further testing in “robo” tasks. A detailed description and classification of “base” tasks as well as an example of their use are available in Appendix, Sections C and I. The set of “base” tasks is needed for initial debugging and testing of memory mechanisms, because “base” tasks are extremely lightweight, simple and fast.
>
> **3. Table 6**
>
> The reference you mentioned in the text should lead to Table 7 in the appendix with the list and classification of tasks from the “base” benchmark. At the same time, Table 6 in the appendix contains detailed results for Offline RL baselines on “robo” tasks. Thank you for finding a typo, we will fix it in the final version.
>
> **4. Intercept task**
>
> Yes. The idea behind Intercept problems is that if the robot only has access to the current camera image at any given time, it will not be able to estimate the velocity (hence the direction of the ball), and thus will not be able to guide the gripper to the correct position for interception. This problem can naturally be solved using frame stacking (as in DQN on Atari tasks), but without it, the robot needs memory to store information about past positions of the ball. Experimental results on the Intercept task are available in Appendix, Section F, Figures 8, 9, 10 and Table 6.
>
> **5. Tasks difficulty**
>
> **5.1. Atomic tasks**
>
> We intentionally isolate individual memory faculties so that failure modes are interpretable. Highly realistic benchmarks (e.g. BEHAVIOR-1k [1]) entangle perception and navigation errors with memory, masking the root cause.
>
> **5.2. Shell Game**
>
> The task “shuffle the cups...” proposed by the Reviewer, just like the one implemented in the ShellGame, essentially tests object permanence, although it is a complicated version of ShellGame. Nevertheless, we find this shuffle version colorful and interesting and will implement it in the next updates of the benchmark, because the time for the rebattle is very limited, although we consider it only as a complication of ShellGame, which is already challenging enough for most algorithms.
>
> **5.3. Long Horizons**
>
> We purposely fixed the configurations of environments with our medium horizons (60-180 timesteps), as this is sufficient to test the fundamental memorization abilities of the agent (if the robot remembers in 1000 steps, it should remember in 10 steps as well), and this medium horizon scenario is already challenging for most baselines. We agree that it can be interesting to test algorithms on custom long horizons, so in the final version of the paper we will add instructions on how to quickly and conveniently configure all memory-related parameters (episode length, event duration, etc.) in our API.
>
> Moreover, for researchers interested in evaluating agent performance on long-horizon tasks, all memory-related parameters in our environments can be easily adjusted out of the box. Below, we provide a concrete example for the RememberColor task, where we increase the episode length from 60 to 1000 steps, extend the visibility duration of the target cube from 5 to 50 steps, and introduce a delay of 900 steps before the cube reappears:
>
> ```python
> from mani_skill.utils.registration import register_env
> from mikasa_robo_suite.remember_color import RememberColorBaseEnv
> import gymnasium as gym
>
> @register_env("RememberColor3Debug-v0", max_episode_steps=1000)
> class RememberColorDebugEnv(RememberColorBaseEnv):
> 	COLORS    	= 3 	# 1-9 unique cubes
> 	TIME_OFFSET   = 50   # how long target cube is shown
> 	GOAL_THRESH   = 0.03  # goal threshold
> 	CUBE_HALFSIZE = 0.02  # cubes size
> 	DELTA_TIME	= 900   # empty table duration
>
> env = gym.make("RememberColor3Debug-v0", num_envs=256, obs_mode="rgb", render_mode="all", delta_time=DELTA_TIME)
> ```
>
> **5.4. VLM with large context**
>
> Even with a 1 000-token window, events older than the window are lost (unless some memory mechanism is used), so medium horizons expose the same forgetting behaviour. Customising horizons (see above) allows researchers to probe this systematically.
>
> **5.5. Cluttering**
>
> Object clutter primarily stresses perception, not memory. Still, following your suggestion we have prototyped a ClutterMemory task (text description provided in the next paragraph) and will ship it in a future release.
>
> **6. More difficult task**
>
> More complex tasks (as in BEHAVIOR-1k) do not explicitly isolate memory usage in the robot's performance, as it may fail an episode, e.g., due to navigation or perception problems. That said, memory puzzle tasks allow disentangle fundamental memory use from other problems.
>
> However, based on your suggestion, we have designed an environment with cluttering that explicitly checks for agent memory:
> - A cube of one of three colors appears on the table in front of the robot in one of three positions in one horizontal straight line. This cube is observed for the first 5 steps.
> - Then this cube disappears from the scene and the robot observes an empty table for 5 steps.
> - Afterward, a cube (including the one the agent saw earlier) appears in each of the three positions.
> - The robot randomly moves each cube forward onto the target behind the die.
> - Once all the cubes are positioned on the target, the robot must move the target cube upward.
>
> Unfortunately, due to conference limitations, we are unable to modify papers, project page and other data, as well as attach images, so we can only provide a text description.
>
> **7. Figure 6**
>
> Category labels here denote the color of a sector of the circle. For ease of reading we will put them in a separate legend in the camera ready version. Thank you for the helpful suggestion!
>
> **8. Online RL Spider Chart**
>
> For online RL, learning curves (Fig. 4-5 and App. F) convey more information than a single end-point radar. Adding another spider chart would duplicate content; we prefer to keep the current presentation.
>
> **9. Observation Space**
>
> MIKASA-Robo supports a variety of observation modalities that can be flexibly combined, enabling evaluation under different levels of observability. As described in Section 6.1, the available observation types include `state`, `RGB`, `joints`, `oracle`, and `prompt`. These can be used individually or in combination to suit the requirements of different algorithms and evaluation setups.
>
> **10. Dense Reward**
>
> Due to the time constraints of the rebuttal period, we were only able to run these additional experiments on a representative subset of tasks — specifically, ShellGameTouch, InterceptMedium, and RememberColor3/5/9. We evaluated the performance of RATE, DT, BC, and CQL models in the Offline RL setting using the **dense** reward function, in addition to the **sparse** reward results already reported in the paper.
>
> |                 | RATE       | DT       | BC        | CQL       |
> |-----------------|------------|-----------|-----------|-----------|
> | ShellGameTouch  | 0.97±0.02  | 0.50±0.17 | 0.38±0.03 | 0.02±0.01 |
> | InterceptMedium | 0.39±0.06  | 0.56±0.02 | 0.51±0.03 | 0.04±0.01 |
> | RememberColor3  | 0.68±0.04  | 0.05±0.03 | 0.20±0.04 | 0.00±0.00 |
> | RememberColor5  | 0.11±.0.04 | 0.05±0.03 | 0.13±0.02 | 0.01±0.01 |
> | RememberColor9  | 0.10±0.01  | 0.02±0.02 | 0.15±0.03 | 0.01±0.00 |
>
> In this paper we consider for Offline RL a scenario with sparse reward, since online RL is not able to work efficiently in such a setting.
>
> **11. VLA Direction**
>
> In the “Limitations” section, we explicitly note that, due to computational constraints, we did not perform extensive hyperparameter sweeps for our models. During fine-tuning we relied on LoRA, and for Octo we used the smaller-parameter Octo-small variant. We do not deny that one could potentially improve the raw scores (for example, achieving an ideally trained memory-less agent’s performance of SR = 1/3 on RememberColor3 or SR = 1/5 on RememberColor5), but not significantly higher. Our current results nevertheless reveal the key performance trends of VLA-based models on MIKASA tasks and underscore their unsatisfactory behavior on problems that demand an explicit memory mechanism.
> To further validate our claims, we conducted additional experiments with pi0 and SpatialVLA (K - action chunk size). As shown by the metrics, these models yield results comparable to other architectures lacking an explicit memory mechanism, thereby reinforcing our conclusion that current popular VLA approaches alone struggle on tasks requiring true memory.
>
> |                 | pi0 (K=4)      | SpatialVLA (K=4)       |
> |-----------------|------------|-----------|
> | ShellGameTouch  | 0.33±0.05  | 0.23±0.04 |
> | InterceptMedium | 0.42±0.03  | 0.27±0.04 |
> | RememberColor3  | 0.35±0.04  | 0.27±0.05 |
> | RememberColor5  | 0.22±0.04 | 0.17±0.03 |
> | RememberColor9  | 0.15±0.02  | 0.11±0.03 |
>
>
> Are there any specific types of realistic robotic memory tasks  beyond the box-filling or cup-shuffling examples that you would particularly recommend for inclusion in future benchmark updates? We want to ensure additions reflect practical scenarios while preserving diagnostic clarity.
>
> Once again, we would like to thank the reviewer for the effort spent on our work as well as for the constructive comments. We hope that our response has clarified the understanding of the paper.
>
> [1] Li, Chengshu, et al. "Behavior-1k: A human-centered, embodied ai benchmark with 1,000 everyday activities and realistic simulation." arXiv preprint arXiv:2403.09227 (2024).

---

> > ### Comment · Reviewer_7Tpx · 2025-08-05
> >
> > Thank you for your response! And I didn't realize there wasn't room for paper revisions during the reviewing process, I apologize for asking for various changes that you didn't have time to make. Regardless, I still feel like my main concern remains, which is that the tasks feel like "toy tasks that happen to involve a robot arm" vs realistic manipulation tasks involving memory. And the tasks still feel very simple/trivial - the fact that existing algorithms fail on them speaks to the limitations of existing algorithms, but we should be designing benchmarks to be ready for future, more capable algorithms.

---

> > > ### Author Response · Authors · 2025-08-06
> > >
> > > Thanks for engaging in the discussion. We appreciate your focus on realism; however, we respectfully disagree with several points:
> > >
> > > 1. **The claim that our tasks are “toy” or trivial contradicts empirical results:** State-of-the-art Online RL, Offline RL, and VLA methods consistently fail on these tasks, even with proposed horizons and dense rewards. This indicates that the tasks are in fact challenging and expose real algorithmic limitations
> > >
> > > 2. **The benchmark is atomic by design:** By stripping away navigation and difficulties that may occur with cluttering and occlusions, MIKASA-Robo isolates memory-specific difficulties. This targeted structure is intentional and valuable for **understanding specific failure modes**, especially in contrast to more complex, entangled setups
> > >
> > > 3. **Benchmarks should address current weaknesses, not hypothetical future models:** A benchmark serves to highlight the limitations of current methods while also creating opportunities for improvement. MIKASA-Robo does exactly this by showing that memory remains a key bottleneck even under controlled conditions by proposing memory tasks with different types of memory and different levels of difficulties
> > >
> > > 4. **Scalability is already built in:** Additionally, our API allows users to easily increase task complexity, such as extending horizons, with a single line of code. If any task becomes solvable, difficulty can be scaled up immediately without infrastructure changes.
> > >
> > > In this way, MIKASA-Robo does not fall behind future progress: we believe that our benchmark will support the community to first solve core memory challenges before moving on to more realistic scenarios. We hope we have been able to clarify your concern and look forward to further discussion.

---

> > > > ### Comment · Reviewer_7Tpx · 2025-08-06
> > > >
> > > > well, all we know is that the baselines you tried don't work very well on the tasks presented. but as is well known by now, RL is very difficult to get right. simply applying some out-of-the box algorithms (without hyperparameter optimization?) doesn't prove that the tasks wouldn't be easily solved by a more in-depth attempt to solve them, using existing methods. this isn't a specific criticism of your experiments here, just generally RL is hard and usually takes more than just applying out of the box algos to get good results. this is why i'd like to see tasks that are more like actual real world tasks - because then, if somebody works on those tasks, and finds them easy, then that's great - that means we've solved something actually useful! whereas if someone solves these tasks you've created here, i just don't feel like that really tells us much about progress towards actual real-world tasks, because these tasks are way simpler than anything we actually care about robots doing in the real world.

---

> > > > > ### Author Response · Authors · 2025-08-09
> > > > >
> > > > > We thank the Reviewer for the perspective. While we agree that RL can be challenging to evaluate fairly, our results are not from “out-of-the-box” runs - we conducted targeted hyperparameter searches for each task and Offline / Online RL baseline, including memory-augmented architectures. The fact that diverse state-of-the-art RL, offline RL, and VLA methods still perform poorly under controlled conditions suggests the tasks are far from “simple.” By design, MIKASA-Robo isolates memory demands from other confounders, so solving these tasks would demonstrate genuine progress in memory reasoning - a capability that underpins many real-world robotics scenarios. Moreover, we release baseline code, enabling the community to inspect all hyperparameters, training setups, and evaluation metrics to verify and reproduce our results.

---

### Official Review · Reviewer_E9xo · 2025-07-02

**Rating:** 5
**Confidence:** 4

**Summary:**

The authors introduce MIKASA, a unified benchmark suite for evaluating memory–dependent reinforcement-learning (RL) agents.

- MIKASA-Base harmonises existing partially-observable tasks under a common Gym-style API.
- MIKASA-Robo contributes 32 tabletop-manipulation environments (12 categories) targeting four memory faculties—object, spatial, sequential, capacity.
- A cognitive-inspired taxonomy of memory tasks is proposed.
- Comprehensive experiments with online RL, offline RL and VLA models reveal significant performance gaps, underscoring the benchmark’s difficulty and diagnostic value.

**Dataset Code Accessibility:**

Yes

**Dataset Code Comments:**

Yes

**Ethical Considerations:**

No, there are no or only very minor ethics concerns

**Final Justification:**

The rebuttal sufficiently addresses my concerns. The additional experiments with GTrXL and DMamba broaden the diversity of evaluated approaches, and the benchmark’s scope across multiple RL paradigms is now clear and well-justified. The clarifications on horizons, task customization, and algorithmic coverage strengthen the submission. I see no critical flaws, and I am satisfied with the authors’ responses. The paper remains a solid and well-motivated contribution, with strong potential for adoption and impact within the community.

**Limitations Weaknesses:**

1. Short episode horizons: Most tasks terminate after 60 ~ 180 steps (e.g., RememberColor 60 steps, ShellGame 90 steps, TakeItBack 180 steps). These limits preclude studying truly long-term credit assignment (> 300-500 steps) seen in navigation or household-assistant settings.

2. Limited algorithmic coverage: SOTA memory architectures (SSMs, GTrXL variants, Retrieval-augmented agents) are absent.

3. Simulation-only scope: No real-robot transfer or mobile-manipulation scenes.

4. Lacks ablations tying taxonomy classes to distinct failure modes.

**Strengths Contributions:**

1. Clear conceptual advance: The four-way taxonomy provides a concise lens for comparing memory mechanisms across domains.

2. Breadth \& difficulty: 32 tasks span 12 scenarios; even strong recurrent baselines collapse as memory load grows.

3. Open, reproducible resources: Code, environment definitions and offline datasets are released under a permissive licence with straightforward installation.

4. Thorough empirical study: Online, offline and VLA evaluations are reported with standard-error bars, offering a valuable baseline for future work.

---

> ### Author Rebuttal · Authors · 2025-07-31
>
> We thank the reviewer for the positive assessment and for highlighting MIKASA-Robo’s diagnostic value.
>
> Below we address each point in detail.
>
> **1. Episode horizons**
>
> Our main goal is to isolate the necessity of memory, not long-horizon credit assignment [1] per se. Many tasks (e.g. RememberColor) becomes non-Markovian after a single occlusion step; thus even 60 steps are sufficient to reveal memory failures.
>
> Further, if one is interested in investigating the agent's performance on long horizon tasks, one can actually tune all the memory-related parameters of the environment (this can be easily and conveniently done out of the box, and below we provide an example instruction for the RememberColor task, in which we increased the episode length from 60 to 1000 steps, as well as increased the duration of the target cube presence from 5 to 50 steps and the time delay until the cube is observed from 5 to 900 steps:
>
> ```python
> from mani_skill.utils.registration import register_env
> from mikasa_robo_suite.remember_color import RememberColorBaseEnv
> import gymnasium as gym
>
> @register_env("RememberColor3Debug-v0", max_episode_steps=1000)
> class RememberColorDebugEnv(RememberColorBaseEnv):
> 	COLORS    	= 3 	# 1-9 unique cubes
> 	TIME_OFFSET   = 50   # how long target cube is shown
> 	GOAL_THRESH   = 0.03  # goal threshold
> 	CUBE_HALFSIZE = 0.02  # cubes size
> 	DELTA_TIME	= 900   # empty table duration
>
> env = gym.make("RememberColor3Debug-v0", num_envs=256, obs_mode="rgb", render_mode="all", delta_time=DELTA_TIME)
> ```
>
> Such customization is applicable to any of the MIKASA-Robo benchmark tasks out of the box, and we will add more detailed instructions in the final version.
>
> **2. Algorithmic coverage**
>
> In our benchmarking, we chose the most common baselines in RL (PPO-MLP, TD-MPC2, SAC-MLP, PPO-LSTM, BC, CQL, DT, DP, RATE) to show that their naive use for any task can lead to poor results on tasks with memory. Among them, we include architectures with memory (PPO-LSTM, DT, RATE). In particular, RATE has a mechanism of caching hidden states as in GTrXL.
>
> To directly address your concern, we conducted additional experiments with state-space models (DMamba [2]) and gated recurrent transformers (GTrXL [3]) on the ShellGameTouch, InterceptMedium, and RememberColor3/5/9 tasks in the offline setting.
>
> |                 | RATE      | DT        | BC        | CQL       | DP        | DMamba    | GTrXL     |
> |-----------------|-----------|-----------|-----------|-----------|-----------|-----------|-----------|
> | ShellGameTouch  | 0.92±0.01 | 0.53±0.07 | 0.28±0.01 | 0.16±0.04 | 0.18±0.02 | 0.21±0.02 | 0.80±0.10 |
> | InterceptMedium | 0.09±0.03 | 0.56±0.01 | 0.31±0.14 | 0.03±0.01 | 0.24±0.01 | 0.14±0.07 | 0.64±0.04 |
> | RememberColor3  | 0.65±0.04 | 0.01±0.01 | 0.27±0.03 | 0.29±0.01 | 0.07±0.04 | 0.32±0.03 | 0.39±0.06 |
> | RememberColor5  | 0.13±0.03 | 0.07±0.05 | 0.12±0.01 | 0.15±0.02 | 0.01±0.01 | 0.11±0.02 | 0.19±0.04 |
> | RememberColor9  | 0.09±0.02 | 0.01±0.01 | 0.12±0.02 | 0.15±0.01 | 0.02±0.01 | 0.14±0.00 | 0.16±0.01 |
>
> These results, together with our originals, confirm that memory-centric SSM/Transformer variants still struggle as capacity and sequence complexity grow. We will add this table and discussion to the supplementary material.
>
> **3. Real-robot transfer**
>
> MIKASA-Robo is intended as a first-principles diagnostic: if an agent cannot recall what happened 5 seconds ago in a noise-free simulator, its sim2real performance is unlikely to be satisfactory. We therefore postpone real-robot adaptation to follow-up work and focus on tabletop manipulation, where memory effects can be cleanly isolated. Navigation-oriented memory benchmarks already exist; our suite fills the manipulation gap (see Table 3).
>
> **4. Ablations**
>
> We would like to clarify that our work already includes extensive comparative evaluations across the proposed taxonomy classes (Object, Spatial, Sequential, and Capacity) using a diverse range of architectures in both offline and online RL settings.
> In the offline RL setting, we evaluate:
> Memory-less baselines: Behavioral Cloning (BC), Conservative Q-Learning (CQL);
> Diffusion-based models: Diffusion Policy (DP);
> Transformers without memory: Decision Transformer (DT);
> Transformers with memory: RATE (see Figure 6).
>
> During the rebuttal, we further included two additional memory-based model families:
> GTrXL (a transformer with recurrence)
> DMamba (a state-space model).
>
> These provide additional diversity in model inductive biases and training strategies.
>
> In the online RL setting, we report results for:
> Model-free policy gradient methods: PPO with MLP and LSTM backbones (PPO-MLP, PPO-LSTM)
> Off-policy and model-based methods: Soft Actor-Critic (SAC) and TD-MPC2.
>
> These experiments cover both memory-less and memory-enhanced agents and span multiple algorithmic paradigms (on-policy, off-policy, model-based, sequence modeling, diffusion, etc.).
> Importantly, our goal is not to propose a new method for solving memory-intensive tasks. Rather, we introduce a benchmark suite that enables systematic comparisons of RL agents across diverse and well-categorized memory challenges. As such, we do not perform targeted ablations, but instead provide a testbed for evaluating how different model classes generalize across the four core types of memory demands.
>
>
> We are again grateful to the reviewer for his constructive feedback and hope that our response has brought clarity to the understanding of our work.
>
> [1] Ni, Tianwei, et al. "When do transformers shine in rl? decoupling memory from credit assignment." Advances in Neural Information Processing Systems 36 (2023): 50429-50452.
>
> [2] Ota, Toshihiro. "Decision mamba: Reinforcement learning via sequence modeling with selective state spaces." arXiv preprint arXiv:2403.19925 (2024).
>
> [3] Parisotto, Emilio, et al. "Stabilizing transformers for reinforcement learning." International conference on machine learning. PMLR, 2020.

---

> ### Comment · Reviewer_E9xo · 2025-08-08
>
> The rebuttal sufficiently addresses my concerns.
> The added models (GTrXL, DMamba) broaden the diversity of approaches, and the benchmark’s scope across multiple RL paradigms is clear and well-justified.
> I am satisfied with the clarifications and maintain my recommendation for acceptance.
>
>
> ---

---

### Official Review · Reviewer_yXHP · 2025-07-03

**Rating:** 5
**Confidence:** 3

**Summary:**

This paper presents MIKASA, a benchmark suite aimed at systematically evaluating memory capabilities in robotic manipulation tasks. To highlight the contribution, this paper argue that existing RL memory benchmarks focus on narrow aspects (e.g., navigation, puzzles) and often fail to represent the physical and memory complexities of real-world robotic scenarios.

**Dataset Code Accessibility:**

Yes

**Ethical Considerations:**

No, there are no or only very minor ethics concerns

**Final Justification:**

I think the author has addressed all of my concerns, and I believe that it's a valuable benchmark for the community. I would like to increase my score to accept.

**Limitations Weaknesses:**

1. Although this benchmark contains realistic tasks which has occlusion and squential decisions, it doesn't have been proved can be benfited for real-world robotics experiments. Because of Sim2Real gap, the evaluation in only simulation environments might be limited.

2. Some of the tasks with sequential memory or high-capacity memory are extremely hard, which leads to nearly zero performance in most of the models. The high difficulty may limit capability for algorithm evaluation. Maybe easier sub-tasks, curricula, or progression strategy are helpful.

3. Although this benchmark is more concrete and has more components than MemoryBench, this improvement might be scientifically incremental.

**Strengths Contributions:**

1. This paper introduces clear and intuitive classification of memory challenges into four types: Object memory, Spatial memory, Sequential memory, and Memory capacity, which are draws from both RL and cognitive science.

2. This benchmark contains large-scale realistic manipulation tasks which focus on memory, which includes 32 task from 12 categories.

3. This paper conducts comprehensive experiments with online RL, offline RL, and VLA.

4. Compared with previous benchmark like MemoryBench, it has much more tasks, different memory types, different diffculty, and various training strategy.

---

> ### Author Rebuttal · Authors · 2025-07-31
>
> Thank you for taking the time to read our work. We are glad that you noticed that our MIKASA-Robo benchmark has more tasks with different categories of memory usage and difficulty levels, as well as experiments with different training strategies compared to MemoryBench, which is **concurrent work** with respect to our work.
>
> Below we address each concern in the order raised.
>
> **1. Sim2Real Gap**
>
> Our goal is to isolate and measure the algorithmic ability to use memory before introducing additional confounders occurring in the sim2real transfer.  If a method fails in simulation under controlled visual occlusion and partial observability, there is no reason to expect it to succeed on hardware. Conversely, once an algorithm demonstrates robust memory use in MIKASA-Robo, sim-to-real adaptation techniques (domain randomization, latent adaptation, etc.) can be explored as a separate research stage.
>
> **2. Task customization**
>
> We deliberately include very challenging variants (e.g. BunchOfColors, SeqOfColors, ChainOfColors) so that the benchmark remains useful as memory-centric algorithms advance.
> To prevent hidden task simplification we propose a set of fixed official configurations with different difficulty levels (e.g. RememberColor3/5/9). At the same time, every environment exposes the underlying parameters (number of objects, episode length, timing intervals, etc.) so researchers can control tasks difficulty for debugging or curriculum learning:
>
> So, for example, here's how you can customize the RememberColor task:
>
> ```python
> from mani_skill.utils.registration import register_env
> from mikasa_robo_suite.remember_color import RememberColorBaseEnv
> import gymnasium as gym
>
> @register_env("RememberColor4Debug-v0", max_episode_steps=1000)
> class RememberColorDebugEnv(RememberColorBaseEnv):
> 	COLORS    	= 4 	# 1-9 unique cubes
> 	TIME_OFFSET   = 200   # how long target cube is shown
> 	GOAL_THRESH   = 0.03  # goal threshold
> 	CUBE_HALFSIZE = 0.02  # cubes size
> 	DELTA_TIME	= 100   # empty table duration
>
> env = gym.make("RememberColor4Debug-v0", num_envs=256, obs_mode="rgb", render_mode="all", delta_time=DELTA_TIME)
> ```
>
> Or, for example, the SeqOfColors task:
>
> ```python
> from mani_skill.utils.registration import register_env
> from mikasa_robo_suite.seq_of_colors import SeqOfColorsEnv
> import gymnasium as gym
>
> @register_env("SeqOfColors6Debug-v0", max_episode_steps=1000)
> class RememberShapeAndColorDebugEnv(SeqOfColorsEnv):
> 	COLORS      	= 2 	# 1-9 unique cubes
> 	GOAL_THRESH 	= 0.03  # goal threshold
> 	CUBE_HALFSIZE   = 0.02  # cubes size
> 	SEQUENCE_LENGTH = 2 	# Number of cubes to show in sequence (1-9)
> 	STEP_DURATION   = 15	# Duration to show each cube
> 	EMPTY_DURATION  = 5 	# Duration of empty table
>
> env = gym.make("SeqOfColors6Debug-v0", num_envs=256, obs_mode="rgb", render_mode="all")
> ```
>
> Instructions for the other tasks are similar and will be added in the final version.
>
>
> **3. Comparison with MemoryBench [1]**
>
> First of all, we would like to note once again that MemoryBench is a **concurrent work** (what we specify in the text in L114, 252) with respect to ours. Moreover, MemoryBench offers only three tasks without different difficulty levels, designed to access only one type of memory (spatial memory), while MIKASA-Robo offers 32 tasks divided into 12 categories with different difficulty levels and covering 4 types of memory (object, spatial, sequential, capacity). Also, only binary sparse reward (Success Metric) is available in MemoryBench, which makes it difficult to use RL algorithms for training (see Figure 10 in Appendix of our paper), while MIKASA-Robo provides both dense reward function and binary sparse reward function. In addition, all work [1] with the MemoryBench benchmark focuses on BC and VLA models, while in MIKASA-Robo we benchmark online and offline RL, BC and VLA. Also MemoryBench offers only single environment simulation, while MIKASA-Robo supports parallel environments.
>
> Finally, MIKASA-Robo includes the dataset published on hugging face for all 32 tasks, and the benchmark itself is available for installation via pip and has a permissive license.
>
> Once again, we are grateful to the reviewer for taking the time to review our work and for the insightful comments. We hope that we were able to clarify the reviewer's comments in our response.
>
> [1] Fang, Haoquan, et al. "Sam2act: Integrating visual foundation model with a memory architecture for robotic manipulation." arXiv preprint arXiv:2501.18564 (2025).

---

> > ### Comment · Reviewer_yXHP · 2025-08-02
> >
> > Thanks for your detailed response. I think the author has addressed all of my concerns, and I believe that it's a valuable benchmark for the community. I would like to increase my score to accept.

---

> > > ### Author Response · Authors · 2025-08-06
> > >
> > > We thank you for the time you spent on our work and for your positive feedback. We are glad you found our benchmark valuable for the community.

---

### Author Response · Authors · 2025-08-09

We sincerely thank all reviewers and area chairs for their time, effort, and constructive feedback. We particularly appreciate the acknowledgment of:

- **Clear motivation and strong narrative supported by a well-defined memory taxonomy** (Reviewers yXHP, XNMz, E9xo)

- **Breadth and diversity of tasks spanning all four taxonomy categories** (Reviewers E9xo, 7Tpx)

- **Open-source availability and reproducibility** (Reviewers 7Tpx, XNMz)

- **Thorough empirical evaluation across online RL, offline RL, and VLA models** (Reviewers yXHP, E9xo, 7Tpx)

- **Configurable environments enabling scalable difficulty for debugging, curriculum learning, and horizon control** (Reviewers E9xo, 7Tpx, XNMz)

- **Well-designed manipulation tasks explicitly targeting memory** (Reviewers yXHP, XNMz)

**Main Concerns Addressed**

- **Task difficulty & customization** – Methods for configuring tasks were presented, including adjusting object count, horizon length, and event timings to create easier or more challenging variants.

- **Algorithmic coverage** – Additional experiments with RL/VLA baselines were conducted during the rebuttal, adding DMamba, GTrXL, pi0, and SpatialVLA, all of which showed performance degradation as memory demands increased.

- **Episode horizons** – It was shown that medium horizons already reveal memory failures, while all tasks support adjustable episode lengths, allowing horizons to be configured as needed.

- **Sim2Real gap** – MIKASA-Robo is designed to test memory in controlled simulation environments, with successful methods intended for later adaptation to real robots.

- **Generalization** – Two-level randomization and disjoint seeds were shown to prevent overfitting and can also be used to evaluate generalization, with several tasks already containing distractors.

In the final version, we will incorporate these clarifications, fix minor inconsistencies, expand customization instructions, and refine the framing to better convey MIKASA-Robo’s scope and role.

Sincerely,

The Authors

---

### Decision · Program_Chairs · 2025-09-18

**Decision:**

Reject

**Comment:**

This paper presents a robotic manipulation benchmark for evaluating memory capability of agents. The tasks are designed systematically, drawing from neuroscientific concepts, and experiments are done using online RL, offline RL, and VLA baselines. However, as several reviewers state, the tasks are still a distance away from real-world robotic manipulation tasks. The rebuttal addressed this concern, stating that the tasks are a first diagnostic of agent capabilities. Since the tasks are challenging for the baselines and are well-designed based on a novel rigorous taxonomy of memory capabilities, representing a step in the right direction for real-world applications, I would recommend to accept as a spotlight.


=================== update from PC ====================


As recently advised by legal counsel, the NeurIPS Foundation is unable to provide services, including the publication of academic articles, involving the technology sector of the Russian Federation's economy under a sanction order laid out in Executive Order (E.O.) 14024.

Based upon a manual review of institutions, one or more of the authors listed on this paper submission has ties to organizations listed in E.O. 14024. As a result this paper has been identified as falling under this requirement and therefore must not be accepted under E.O. 14024.

This decision may be revisited if all authors on this paper can provide proof that their institutions are not listed under E.O. 14024 to the NeurIPS PC and legal teams before October 2, 2025. Final decisions will be communicated soon after October 2nd. Appeals may be directed to datasetsbenchmarks@neurips.cc